# Arginine methylation-dependent METTL14-SMN interaction regulates RNA m⁶A homeostasis

Yi Zhang [1], Lei Shen[1], Lili Ren [2], Jiangbo Wei[3,6], Hoang Quoc Hai Pham [1,4], Xiaoqun Tao[1,4], Jiamin Guo [4], Zhihao Wang[1], Binghui Shen [1], Rui Su[2], Chuan He [3,5] & Yanzhong Yang [1,4]✉

## Abstract

N6-methyladenosine (m⁶A) homeostasis is essential for development, and its dysregulation is linked to cancers and neurological disorders. However, the mechanisms regulating m⁶A remain unclear. Here, we identify the survival of motoneuron (SMN) protein as a novel interaction partner of METTL14, a key component of the m⁶A methyltransferase complex. SMN binds METTL14 via its Tudor domain in an arginine methylation-dependent manner. Mutations in the SMN Tudor domain identified in spinal muscular atrophy (SMA) disrupt its interaction with METTL14 and reduce m⁶A levels in patient-derived fibroblasts, linking m⁶A dysregulation to SMA pathology. Both SMN knockdown and SMA mutations impair m⁶A deposition on the mRNAs of DNA repair genes, mirroring the effects of METTL14 hypomethylation. Consequently, SMA patient fibroblasts are hypersensitive to DNA-damaging agents due to reduced levels of DNA repair gene expression. To explore the function of METTL14 arginine methylation in vivo, we generated a *Mettl14* methylation-deficient mouse model (*Mettl14^RK*). Although this model does not show SMA-like phenotypes, the mutants are partially embryonic lethal and show abnormal hematopoiesis, underscoring a role for methylated METTL14 in early development.

**Keywords** Arginine Methylation; METTL14; SMN; Genome Stability; Hematopoiesis
**Subject Categories** Molecular Biology of Disease; Post-translational Modifications & Proteolysis; RNA Biology

## Introduction

Arginine methylation is an abundant post-translational modification (PTM) implicated in the regulation of multiple important biological processes, including gene transcription, DNA repair, mRNA splicing, and signal transduction (Bedford and Clarke, 2009;

Yang and Bedford, 2013). In mammalian cells, there are three types of methylarginine species: ω-N^G-monomethylarginine (MMA), ω-N^G, N^G-asymmetric dimethylarginine (ADMA), and ω-N^G, N^G-symmetric dimethylarginine (SDMA), catalyzed by a family of nine protein arginine methyltransferases (PRMT1–9) (Bedford and Clarke, 2009; Yang and Bedford, 2013). Based on the methylation products that they produce, mammalian PRMTs can be classified into three catalytic groups: type I PRMTs that produce MMA and ADMA include PRMT1, PRMT2, PRMT3, PRMT4/CARM1, PRMT6, and PRMT8; type II PRMTs that produce MMA and SDMA include PRMT5 and PRMT9, and type III PRMT PRMT7, which only produces MMA. Arginine methylation could potentially alter protein structures, impact protein–DNA/RNA integrations, and generate docking sites for effector proteins. Currently, the Tudor domain-containing proteins are the "primary" readers of methylarginine modifications (Chen et al, 2011). These domains are ~60 amino acids in size and use conserved aromatic residues to build an 'aromatic cage' and bind methylated-arginine through cation–π and π–π stacking interactions. A majority of the methylarginine-binding Tudor domain proteins show enriched expression in germ cells and play evolutionarily conserved roles in germinal granule/nuage formation and germ cell specification (Lasko, 2010; Pek et al, 2012). The Tudor domain-containing protein 3 (TDRD3) and the human survival motor neuron (SMN) are two of the ubiquitously expressed methylarginine effectors involved in many of the arginine methylation-mediated pathways (Espejo and Bedford, 2015; Goulet et al, 2008; Narayanan et al, 2017; Pek et al, 2012; Selenko et al, 2001; Yang et al, 2010; Yang et al, 2014; Yuan et al, 2021; Zhao et al, 2016). Elevated expression of TDRD3 has been associated with poor survival of ER-negative and basal-like breast cancer patients (Hallett et al, 2012; Morettin et al, 2017; Nagahata et al, 2004), whereas genetic mutation of SMN1 gene causes human spinal muscular atrophy (SMA) syndrome, a genetic disorder that results in a loss of motor neurons (Crawford and Pardo, 1996; Kariya et al, 2008; Lefebvre et al, 1995), highlighting the important roles of methylarginine effector molecules.

N6-methyladenosine (m⁶A) is the most abundant internal modification of mRNA and involves in every step of mRNA life cycle, including splicing, translation, and stability (Boulias and

[1]Department of Cancer Genetics and Epigenetics, Beckman Research Institute, City of Hope, Duarte, CA, USA. [2]Department of System Biology, Beckman Research Institute, City of Hope, Duarte, CA, USA. [3]Department of Chemistry, Department of Biochemistry and Molecular Biology, and the Institute for Biophysical Dynamics, The University of Chicago, Chicago, IL 60637, USA. [4]Irell & Manella Graduate School of Biological Sciences, Beckman Research Institute, City of Hope, Duarte, CA, USA. [5]Howard Hughes Medical Institute, University of Chicago, Chicago, IL 60637, USA. [6]Present address: Department of Chemistry and Department of Biological Sciences, National University of Singapore, Singapore 117544, Singapore. ✉E-mail: yyang@coh.org

Greer, 2023; Zaccara et al, 2019). A majority of m⁶A modifications on mRNA are catalyzed by a multicomponent methyltransferase complex containing the methyltransferase-like 3 (METTL3)/methyltransferase-like 14 (METTL14) heterodimer and other regulatory factors. Although genetic knockout studies of METTL3 and METTL14 have provided valuable information in understanding the role of m⁶A-mediated mRNA metabolism in development and human diseases (Hsu et al, 2017; Jaffrey and Kharas, 2017; Pan et al, 2018; Vu et al, 2017; Weng et al, 2018; Yoon et al, 2017), the molecular mechanisms underlying the regulation of m⁶A are largely unknown. Recently, we and others reported that arginine methylation is a critical post-translational modification (PTM) that regulates METTL14 function and m⁶A deposition (Liu et al, 2021; Wang et al, 2023a; Wang et al, 2023b; Wang et al, 2021). Importantly, METTL14 arginine methylation-dependent m⁶A sites are specifically enriched in genes involved in interstrand crosslink (ICL) repair pathway, thus cells expressing arginine methylation-deficient METTL14 are hypersensitive to DNA crosslinking agents (Wang et al, 2021).

Here, to further dissect the molecular mechanisms by which arginine methylation of METTL14 promotes m⁶A deposition, we seek to determine the contribution of methylarginine effectors in the regulation of m⁶A deposition. Using biochemical and cellular assays, we show that SMN interacts with METTL14 through its Tudor domain in an arginine methylation-dependent manner. This interaction enhances METTL14 interaction with RNA polymerase II (RNAPII). HeLa cells with knockdown expression of SMN and fibroblast cells derived from patients with SMA exhibited reduced levels of m⁶A in mRNA. Consistent with the role of METTL14 arginine methylation in promoting m⁶A deposition on ICL DNA repair genes, both knockdown SMN and SMA-derived SMN mutations cause reduced levels of m⁶A deposition on ICL gene transcripts and protein expression, which are associated with their increased sensitivity to DNA damage agents. Furthermore, to define the biological function of METTL14 arginine methylation in vivo, we established a *Mettl14* arginine methylation-deficient mutant mouse model. Although we did not observe SMA-like phenotypes, the homozygous mutant mice exhibited partial embryonic lethality and abnormal hematopoiesis, akin to *Mettl14* knockout (KO) mice. Altogether, these results demonstrate a previously unrecognized relationship between impaired m⁶A homeostasis and the pathogenesis of SMA, shedding light on the crucial role of arginine methylation in regulating m⁶A and its broader implications for developmental biology.

# Results

## SMN interacts with METTL14 through its methylarginine binding Tudor domain

Methylation signals on the arginine residues are often recognized by the Tudor domain containing proteins, known as methylarginine readers. Among them, SMN and TDRD3 are two major ubiquitously expressed readers involved in transcription regulation and RNA metabolism. To further characterize the biological function of METTL14 arginine methylation, we tested the hypothesis that arginine methylation could promote the interactions between METTL14 and the methylarginine reader(s). To do

this, we performed GST pull-down assays using the recombinant Tudor domain proteins of TDRD3 and SMN, along with their respective methylarginine binding deficient mutants, E691K and E134K, respectively (Cote and Richard, 2005; Tripsianes et al, 2011). The Tudor domain of SMN, but not its methylarginine binding deficient E134K mutant or the Tudor domain of TDRD3, was able to pull down three components of the m⁶A methyltransferase complex, including METTL14, METTL3, and WTAP (Fig. 1A). To determine which of the three components is responsible for this interaction, we performed a co-immunoprecipitation (co-IP) assay in HeLa cells using Flag-tagged METTL14, METTL3, and WTAP. SMN was detected to mainly interact with METTL14 (Fig. 1B). To further confirm this interaction, we performed a reciprocal co-IP assay on the endogenously expressed METTL14 and SMN. Indeed, endogenous METTL14 and SMN interacted with each other (Fig. 1C,D). Treatment of cell lysates with RNase A does not affect the interaction of SMN Tudor domain with METTL14 (Appendix Fig. S1A), suggesting that their interaction is not mediated by RNA. We also assessed their colocalization in cells using the immunofluorescence assay and observed that these two proteins colocalize in the nucleus, with strong signals likely within Cajal bodies, where SMN is known to localize (Liu and Dreyfuss, 1996) (Appendix Fig. S1B). Additionally, we performed a co-IP assay with Flag-tagged WT or Tudor domain E134K mutant SMN, and found that, consistent with the pull-down results (Fig. 1A), E134K mutant SMN failed to interact with METTL14 (Fig. 1E), suggesting that their interaction relies on the methylarginine binding ability of the SMN Tudor domain. SMA is a common motor neuron disease that results from loss-of-function mutations in the SMN1 gene. Although the vast majority of SMA cases result from a large deletion in SMN1 (Crawford and Pardo, 1996; Kariya et al, 2008; Lefebvre et al, 1995), 1% of SMA cases are caused by loss-of-function missense mutations, some of which are located in the Tudor domain (Cusco et al, 2004; Kotani et al, 2007; Selenko et al, 2001; Takarada et al, 2017). SMA-derived mutations located in the Tudor domain of SMN reduce the interaction between SMN and METTL14, supporting the functional role of the Tudor domain in mediating its interaction with METTL14 (Appendix Fig. S1C,D). Altogether, these results revealed SMN as a new interaction partner of METTL14.

## METTL14 interacts with SMN in an arginine methylation-dependent manner

Tudor domains often mediate protein–protein interactions in an arginine methylation-dependent manner (Chen et al, 2011). Based on our previous report that METTL14 is heavily arginine methylated at its C-terminal intrinsically disordered region (Wang et al, 2021), we hypothesized that arginine methylation is essential for the interaction between METTL14 and SMN. To test this hypothesis, we first performed a GST pull-down experiment by incubating recombinant Tudor domain of SMN with HeLa cell lysates transfected with Flag-tagged full length (FL) or C-terminus truncated (1-400) METTL14. Removing the arginine methylated C-terminus abolished the interaction of METTL14 and SMN (Fig. 2A). To test if this interaction depends on METTL14 arginine methylation, we performed a similar GST pull-down experiment by incubating recombinant SMN Tudor domain with HeLa cell lysates

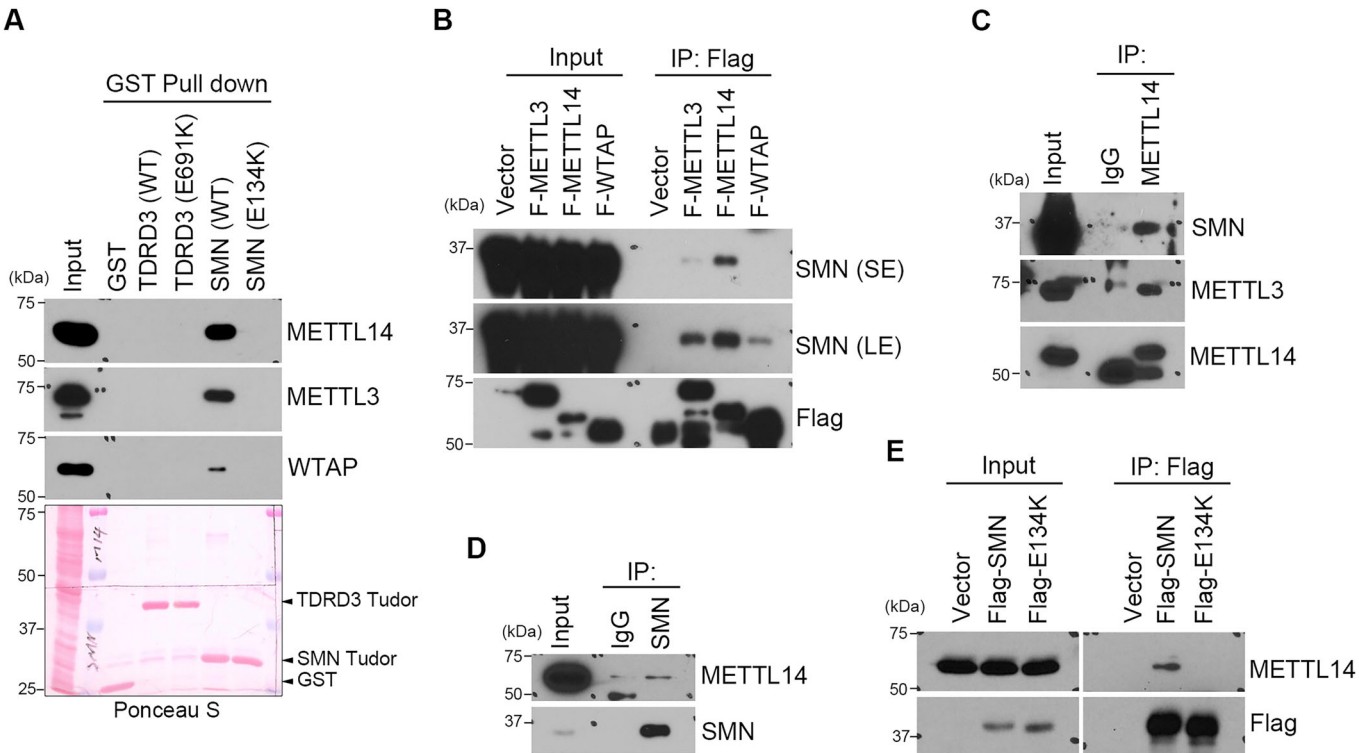

**Figure 1. SMN interacts with METTL14 through its Tudor domain.**

(A) Tudor domain of SMN interacts with the m6A methyltransferase complex. GST pull-down was performed by incubating HeLa cell lysates with wild-type (WT) and methylarginine binding deficient mutant Tudor domains of TDRD3 and SMN. The amount of recombinant proteins used for the pull down was detected by ponceau staining of the membrane. (B) SMN preferentially interacts with METTL14. HeLa cells were transfected with Flag-METTL3, Flag-METTL14, and Flag-WTAP. A co-immunoprecipitation (co-IP) assay was performed to detect their interaction with SMN. SE short exposure, LE long exposure. (C) Endogenous SMN interacts with METTL14. Endogenous co-IP was performed using the IgG control or METTL14 antibodies for IP and the SMN antibody for western blot detection. (D) Endogenous METTL14 interacts with SMN. Endogenous co-IP was performed using the IgG control or SMN antibodies for IP and the METTL14 antibody for detection. (E) Methylarginine binding deficient SMN failed to interact with METTL14. A co-IP assay was performed to detect the interaction of METTL14 with either WT or methylarginine binding deficient mutant (E134K) Flag-tagged SMN. Source data are available online for this figure.

expressing WT and various arginine methylation-deficient METTL14 mutants, namely 5RK: R438/442/445/450/456K; 8RK: R429/431/435/438/442/445/450/456K; 10RK: R425/427/429/431/435/438/442/445/450/456K; and 13RK: R408/414/418/425/427/429/431/435/438/442/445/450/456K (Wang et al, 2021). Consistent with our previous observation that the 5RK mutation is sufficient to cause dramatic arginine methylation loss on METTL14, it consequently disrupted METTL14 interaction with the SMN Tudor domain (Fig. 2B). Next, we performed a co-IP experiment using HeLa cell lysates transfected with Flag-tagged full length (FL), arginine methylation-deficient (RK) mutant, and C-terminus truncated (1-400) mutant METTL14. SMN only interacted with wild-type (WT) METTL14 (Fig. 2C), suggesting that their interaction is arginine methylation dependent. Supporting the role of PRMT1 being the major PRMT that methylates METTL14 (Wang et al, 2023a; Wang et al, 2021), knockdown of PRMT1 expression using siRNA significantly reduced the interaction of METTL14 with SMN as revealed by both co-IP and GST pull down assays (Fig. 2D,E). To further confirm that arginine methylation is directly involved in METTL14–SMN interaction, we inhibited cellular arginine methylation using two PRMT inhibitors, MS023

(Eram et al, 2016), which targets the majority of the type I PRMTs (ADMA) and EPZ015666 (EPZ) (Chan-Penebre et al, 2015) that specifically inhibits PRMT5 (SDMA) either individually or in combination. MS023 treatment dramatically reduced the level of METTL14 arginine methylation as detected by the pan-ADMA antibody, and, as a result, its interaction with SMN was also reduced (Fig. 2F). Surprisingly, although the SDMA modification on METTL14 was only weakly detected, its level increased significantly upon MS023 treatment, consistent with the general observation of PRMT substrate scavenging, meaning that when type I PRMTs are inhibited, their native substrates become available for methylation by PRMT5 (Dhar et al, 2013). In line with this conclusion, although EPZ treatment alone has marginal effects on the METTL14–SMN interaction, combinatory treatment using both MS023 and EPZ greatly reduced their interaction when compared to the MS023 treatment alone (Fig. 2F), suggesting that both ADMA and SDMA contribute to METTL14–SMN interaction. This observation was also independently validated by using a GST pull down assay (Fig. 2G). To further characterize the significance of SDMA modification in the METTL14–SMN interaction, we observed that PRMT5 only weakly methylates METT14 in vitro

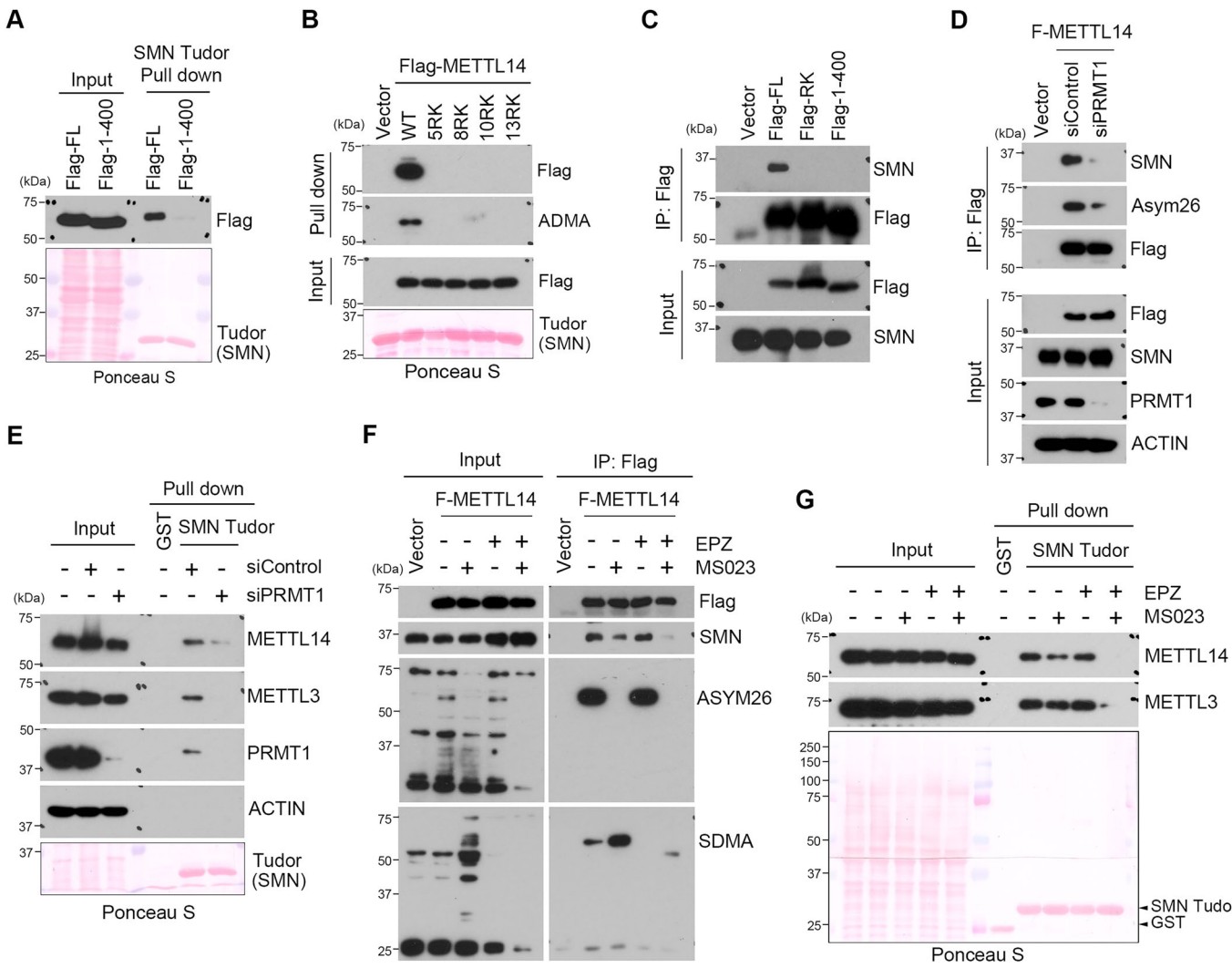

**Figure 2. SMN interacts with METTL14 in an arginine methylation-dependent manner.**

(A) METTL14 C-terminal disordered region is essential for its interaction with the Tudor domain of SMN. GST pull-down was performed to detect the interactions between the Tudor domain of SMN and either Flag-tagged full length (FL) or C-terminal deleted (1–400) METTL14. (B) Arginine methylation deficient METTL14 fails to interact with SMN Tudor domain. GST pull-down was performed to detect the interactions between the recombinant Tudor domain of SMN and either WT or various R-to-K mutant Flag-tagged METTL14. (C) Arginine methylation sites located at the C-terminal disorder region of METTL14 are essential for its interaction with SMN. A co-IP assay was performed to detect the interactions of SMN with Flag-tagged WT, arginine methylation-deficient mutant (RK), and C-terminus deleted (1–400) METTL14. (D) PRMT1-catalyzed METTL14 arginine methylation is critical for its interaction with SMN. A co-IP assay was performed to detect the interactions of Flag-METTL14 with SMN in control (siControl) and PRMT1 knockdown (siPRMT1) HeLa cells. The methylation status of Flag-tagged METTL14 was detected by using the pan-ADMA (Asym26) antibody. (E) PRMT1 knockdown reduces the interaction of SMN with the m6A methyltransferase complex. GST pull down was performed by incubating cell lysates from control siRNA (siControl) or PRMT1-specific siRNA (siPRMT1) transfected HeLa cells with the recombinant Tudor domain of SMN. (F) SMN interacts with METTL14 in an arginine methylation-dependent manner. A co-IP assay was performed to detect the interactions of Flag-METTL14 with SMN in HeLa cells treated with type I and type II PRMT inhibitor, MS023 and EPZ (EPZ015666), either individually or in combination. The methylation status of METTL14 was detected by using the pan-ADMA (ASYM26) and the pan-SDMA antibodies. (G) Tudor domain of SMN interacts with METTL14/METTL3 complex in an arginine methylation-dependent manner. GST pull-down was performed by incubating the recombinant Tudor domain of SMN with HeLa cell lysates that were treated with MS023 and EPZ (EPZ015666), either individually or in combination. The amount of recombinant proteins used for the pull down was detected by ponceau staining of the membrane. Source data are available online for this figure.

(Appendix Fig. S2A), and that PRMT5 knockdown using siRNA did not significantly affect their interaction (Appendix Fig. S2B). Therefore, the contribution of PRMT5-mediated SDMA modification of METTL14 is likely only evident when type I PRMT activity is inhibited. Altogether, these results strongly support that METTL14 interacts with SMN in an arginine methylation-dependent manner.

## SMN facilitates the association of METTL14 with RNAPII

Arginine methylation of METTL14 has been shown to increase its interaction with RNA substrate and enhance its association with RNAPII (Wang et al, 2021). Given that SMN can also interact with RNAPII complex (Zhao et al, 2016), we hypothesized that SMN–METTL14 interaction facilitates the association of METTL14

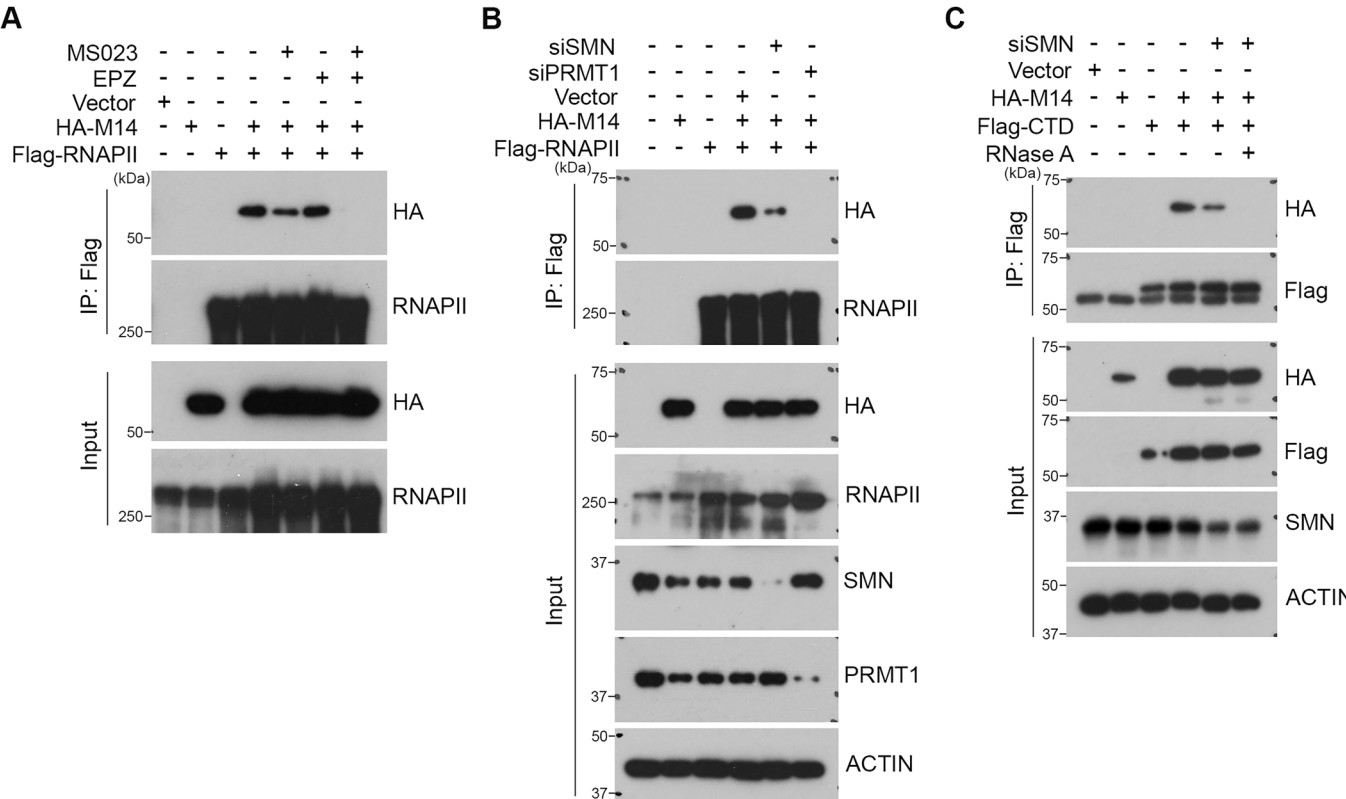

**Figure 3. SMN regulates METTL14 interaction with RNAPII.**

(A) METTL14 interacts with RNAPII in an arginine methylation-dependent manner. A co-IP assay was performed to detect the interaction of HA-tagged METTL14 with Flag-tagged RNAPII in HeLa cells treated with MS023 and EPZ (EPZ015666), either individually or in combination. (B) Both PRMT1 and SMN are important for the interaction of METTL14 and RNAPII. A co-IP assay was performed to detect the interaction of HA-tagged METTL14 with Flag-tagged RNAPII in HeLa cells with knockdown expression of either SMN or PRMT1. (C) SMN facilitates the interaction of METTL14 with the C-terminal domain (CTD) of RNAPII. A co-IP assay was performed to detect the interaction of HA-tagged METTL14 with Flag-tagged RNAPII CTD in control and SMN knockdown HeLa cells. Treatment with RNase A further reduces the interaction of METTL14 with RNAPII CTD in SMN knockdown cells. Source data are available online for this figure.

with transcribing RNAPII. To test this, we first confirmed the potential involvement of both ADMA and SDMA modifications in promoting METTL14–RNAPII interaction. Indeed, combinatory treatment of cells using MS023 and EPZ dramatically reduced their interaction (Fig. 3A), further demonstrating that this interaction is sensitive to loss of arginine methylation. Next, to define the role of SMN in this process, we assessed METTL14–RNAPII interaction in HeLa cells transfected with siRNA targeting either SMN or PRMT1, with the latter serving as a positive control. Reducing SMN expression significantly dampened the interaction of METTL14 and RNAPII (Fig. 3B). Interestingly, although the expression of SMN was strongly reduced by siRNA knockdown (>90%), its effect on the METTL14–RNAPII interaction was less pronounced than that observed with PRMT1 knockdown (Fig. 3B, compare Lane 5 with Lane 6), indicating that SMN only partially mediates the functional outcome of METTL14 arginine methylation. We previously reported that arginine methylation of METTL14 enhances its interaction with RNA substrates (Wang et al, 2021). To test if RNA-mediated interactions contribute to the residual METTL14–RNAPII association after SMN knockdown, we treated cell lysates with RNase A to remove potential RNA mediated protein–protein interactions and observed that combination of

SMN knockdown and RNase A treatment abolished the METTL14–RNAPII interaction (Fig. 3C). Together, these results support a role for SMN in facilitating the association of METTL14 with RNAPII.

## SMN contributes to cellular m⁶A maintenance

The interaction of SMN with METTL14 and the impact of SMN loss on METTL14–RNAPII interaction led us to hypothesize that SMN plays a role in maintaining cellular m⁶A homeostasis. To test this, we first knocked down the expression of SMN in HeLa cells using siRNA (Fig. 4A) and purified polyadenylated mRNA from these cells. Next, we compared the levels of mRNA m⁶A using two independent approaches, namely m⁶A dot-blot and liquid chromatography–tandem mass spectrometry (LC-MS/MS). As shown in Fig. 4B,C, knockdown of SMN expression caused a mild (about 17%), but statistically significant global reduction of m⁶A on mRNAs. Loss-of-function mutation of the survival of motor neuron 1 (SMN1) gene, which results in insufficient levels of SMN protein, is the causes of the neuromuscular disease, SMA. Most humans possess at least one copy of an additional the SMN2 gene, which is almost identical to SMN1. Thus, the copy number of SMN2

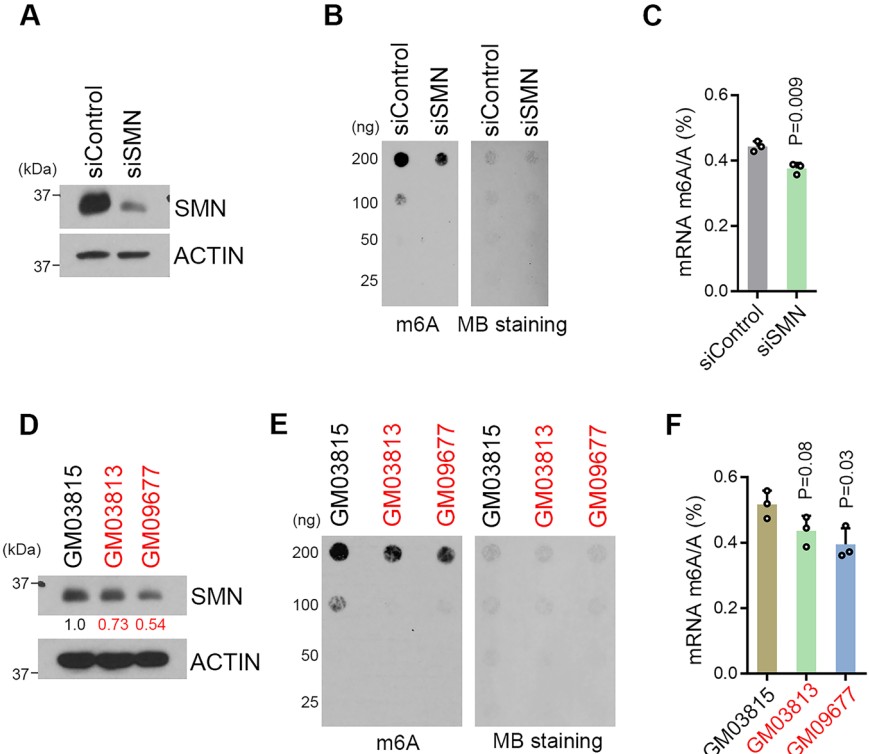

**Figure 4. SMN is involved in the maintenance of cellular m⁶A level.**

(A) Western blot detection of SMN protein levels in control siRNA (siControl) and SMN-specific siRNA transfected HeLa cells. (B) Dot blot detection of m⁶A levels of the mRNA samples purified from siControl and siSMN transfected HeLa cells. (C) SMN knockdown reduces global m⁶A levels in mRNA in HeLa cells. The mRNA purified from HeLa cells transfected with control siRNA (siControl) and SMN-specific siRNA (siSMN) was subjected to LC-MS/MS analysis to quantify m⁶A levels (presented as the m⁶A/A ratio). Data from three independent replicates were analyzed by Student's *t* test and shown as mean ± SD. (D) Western blot detection of SMN protein levels in SMA patient-derived fibroblasts. Fibroblast cells from the two clinically affected individuals were labeled in red. (E) Dot blot detection of m⁶A levels of the mRNA samples purified from SMA patient-derived fibroblasts. (F) m⁶A levels are reduced in SMA patient-derived fibroblast. The mRNA purified from three SMA patient-derived fibroblasts was subjected to LC-MS/MS analysis to quantify m⁶A levels (presented as the m⁶A/A ratio). Data from three independent replicates were analyzed by Student's *t* test and shown as mean ± SD. Source data are available online for this figure.

contributes significantly to the disease severity, with an increased SMN2 copy number generally predicting a less severe SMA phenotype (Cusco et al, 2020; Gavrilov et al, 1998). To further confirm the functional involvement of SMN in cellular m⁶A maintenance, we compared the m⁶A levels in three SMA patient-derived fibroblast cells. GM03815 is from a clinically unaffected male who is a carrier for SMN1 and has one copy of the SMN2 gene. He is the father of the clinically affected donor for GM03813, who is homozygous for the deletion of exons 7 and 8 in the SMN1 gene and has 3 copies of the SMN2 gene (Stabley et al, 2017; Wan et al, 2005). GM09677 is from a clinically affected 2-year-old who is homozygous for deletion of both exons 7 and 8 of the SMN1 gene and has 3 copies of the SMN2 gene (Stabley et al, 2015). Based on their clinical features and the dosage of the SMN2 gene, GM03815 and GM03813 are classified to SMN II (less severe form of SMA with later onset), and GM09677 was classified to SMA I (the most severe form of SMA with an early onset in infancy) (Stabley et al, 2015). The SMN protein expression in GM03813 is slightly lower than in the GM03815, whereas GM09677 has the lowest SMN protein expression (Fig. 4D). Correlated with their respective SMN protein level, both m⁶A dot-blot and LC-MS/MS detected reduced m⁶A levels in clinically affected SMA patient fibroblast cells, with

GM09677 (most severe form of SMA) being the lowest (Fig. 4E,F). Altogether, these results demonstrated that SMN is functionally involved in the regulation of cellular m⁶A homeostasis.

## SMN promotes DNA repair gene expression

Previously, we reported that the m⁶A deposition on DNA repair genes, especially those involved in the interstrand crosslink (ICL) repair, is specifically sensitive to the loss of PRMT1-catalyzed METTL14 arginine methylation (Wang et al, 2021). The reduction of global mRNA m⁶A upon SMN knockdown (Fig. 4) has led us to test if SMN is involved in m⁶A deposition on DNA repair genes. To test this, we performed the methylated RNA (m⁶A) immunoprecipitation (MeRIP) on HeLa cells treated with PRMT inhibitor MS023 or with knockdown expression of SMN. Detection of m⁶A levels on those DNA repair genes using RT-qPCR revealed a mild but statistically significant reduction upon PRMT inhibition or SMN loss (Fig. 5A). Similarly, SMA patient-derived fibroblast cells also exhibited reduced m⁶A deposition on DNA repair genes (Fig. 5B). Note that knockdown of SMN does not have significant effects on either the mRNA or protein levels of key m⁶A pathway components, such as writers, readers, and erasers of m⁶A

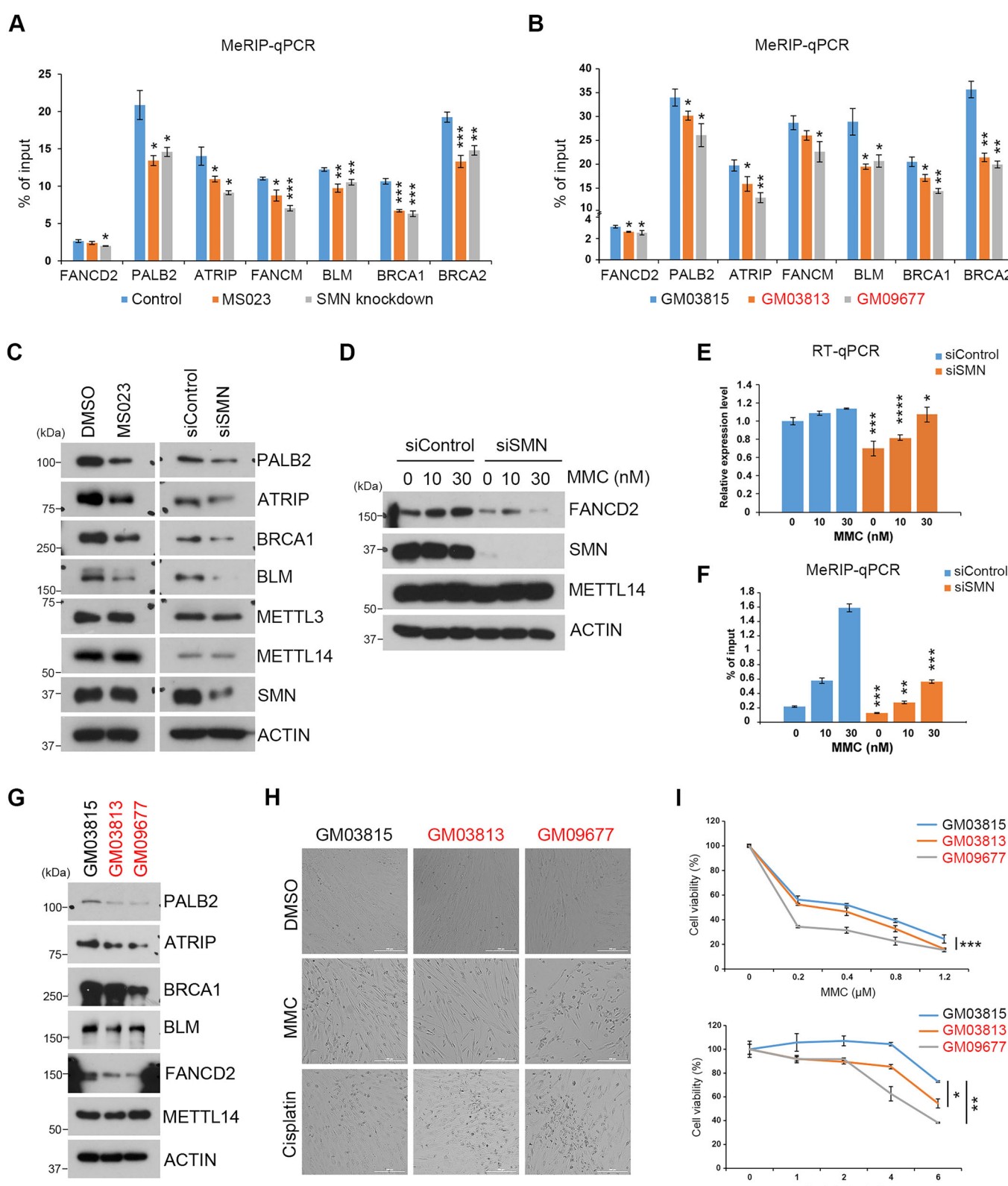

(Appendix Fig. S3A,B). Although we cannot rule out the possibility that other molecular pathways affected by SMN loss may contribute to the changes in m⁶A deposition, the splicing of these DNA repair genes around the m⁶A modified exons were not significantly altered by SMN knockdown (Appendix Fig. S3C). Supporting the role of

m⁶A in facilitating DNA repair gene mRNA translation efficiency, knockdown of SMN or inhibiting PRMT1 activity reduced the protein levels of these DNA repair genes (Fig. 5C), consistent with our previous observation in mESCs (Wang et al, 2021). Additionally, we observed a synergistic effect of co-treatment with MS023

**Figure 5. SMN-regulated m⁶A deposition is critical for the expression of DNA repair genes and cellular DNA damage response.**

(A) Inhibiting PRMT activity or knockdown of SMN expression reduces m⁶A deposition on DNA repair genes. MeRIP-qPCR was performed to detect the levels of m⁶A on the transcripts of several DNA repair genes. RNA was extracted from control, MS023-treated, and SMN knockdown HeLa cells followed by immunoprecipitation using an m⁶A antibody. *P = 0.0151, *P = 0.0144, *P = 0.0225, *P = 0.0386, *P = 0.0165, *P = 0.0272, ***P = 0.0005, **P = 0.0073, **P = 0.0047, ***P = 0.0008, ***P = 0.0001, ***P = 0.0007, **P = 0.0012, from left to right. (B) The levels of m⁶A on the transcripts of several DNA repair genes were reduced in SMA patient-derived fibroblasts. Similar to (A), RNA extracted from SMA patient-derived fibroblasts was subjected to MeRIP-qPCR analysis. *P = 0.0131, *P = 0.0225, *P = 0.0466, *P = 0.0125, *P = 0.0285, **P = 0.0018, *P = 0.0194, *P = 0.0245, *P = 0.0217, *P = 0.0110, **P = 0.0025, **P = 0.0011, **P = 0.0013, from left to right. (C) Inhibiting PRMT activity or knockdown of SMN expression reduces the protein levels of several DNA repair genes. Western blot analysis was performed on total cell lysates from HeLa cells treated with MS023 or transfected with siRNA targeting SMN. (D) Knockdown of SMN dampens Mitomycin C (MMC)-induced FANCD2 expression. HeLa cells were transfected with either control siRNA (siControl) or SMN-specific siRNA (siSMN) and subjected to a dosage-dependent MMC treatment for 72 h. The total cell lysates were subjected to western blot analysis using indicated antibodies. (E) Knockdown of SMN has a marginal effect on FANCD2 mRNA expression. The RNA expression of FANCD2 was analyzed in HeLa cells treated as described in (D) by using RT-qPCR. ***P = 0.0008, ****P < 0.0001, *P = 0.0168, from left to right. (F) Knockdown of SMN dampens MMC-induced m⁶A deposition on FANCD2 transcripts. MeRIP-qPCR was performed to compare the levels of m⁶A on FANCD2 in HeLa cells treated as described in (D). ***P = 0.0006, **P = 0.0014, ***P = 0.0002, from left to right. (G) The protein levels of several DNA repair genes were reduced in SMA patient-derived fibroblasts. Western blot analysis was performed on total cell lysates from SMA patient-derived fibroblasts. (H) SMN loss sensitizes fibroblasts to DNA damage induced cell death. MMC and Cisplatin induced cell death on SMA patient-derived fibroblasts were visualized by bright field microscopy. (I) Similar to (H), the viability of SMA patient-derived fibroblasts were quantitively measured using CCK-8 assay under increasing dosage of MMC and Cisplatin. In (A, B, E, F), data from three independent replicates were analyzed by Student's t test and shown as mean ± SD. ***P = 0.0002, *P = 0.0133, **P = 0.0040, from top to bottom. Source data are available online for this figure.

and EPZ in reducing m⁶A deposition (Appendix Fig. S3D) and DNA repair gene protein expression (Appendix Fig. S3E) compared to MS023 alone, further supporting the compensatory role of PRMT5-mediated METTL14 SDMA modification during MS023 treatment. We noticed that the protein level of Fanconi anemia complementation group D2 (FANCD2) is induced when cells are treated with increasing amounts of Mitomycin C (MMC), an ICL inducing agent (Fig. 5D). This induction was markedly reduced in cells with SMN knockdown (Fig. 5D), suggesting that SMN is essential for MMC-induced upregulation of FANCD2. Importantly, knockdown METTL14 (Appendix Fig. S4A) or inhibiting m⁶A methyltransferase activity using a METTL3-specific inhibitor (Appendix Fig. S4B) both dampened MMC-induced upregulation of FANCD2, supporting an m⁶A-dependent mechanism. The mRNA expression level of FANCD2 remains largely unchanged upon MMC treatment (Fig. 5E), but rather the m⁶A level on its RNA transcript was strongly induced and, importantly, this induction of m⁶A was diminished in cells with knockdown expression of SMN (Fig. 5F), indicating that MMC-induced upregulation of FANCD2 protein expression likely involves elevated m⁶A deposition in an SMN-dependent manner. Although it remains unclear why other m⁶A-targeted DNA repair genes do not behave this way, we did notice that the level of m⁶A on FANCD2 transcript is five- to tenfold lower than that on other DNA repair gene transcripts in both HeLa and SMA patient-derived fibroblasts (Fig. 5A,B), suggesting that increasing DNA repair protein expressing through enhanced m⁶A deposition could be a unique mechanism by which cells elevate DNA repair capacity upon damage. Consistent with reduced levels of DNA repair gene expression, HeLa cells with knockdown expression of SMN are more sensitive to cell death induced by ICL damaging agents, including MMC and Cisplatin (Appendix Fig. S4C,D).

Next, we examined DNA repair gene expression in SMA patient-derived fibroblast cells and found that cells from two clinically affected individuals showed reduced DNA repair gene expression (Fig. 5G), attenuated MMC-induced FANCD2 expression (Appendix Fig. S4E) and decreased m⁶A deposition (Appendix Fig. S4F,G), and consequently increased sensitivity to DNA damage-induced cell death (Fig. 5H,I). Altogether, these results link the function of SMN with DNA repair gene expression, likely through m⁶A-mediated pathway.

## METTL14 arginine methylation deficiency affects normal hematopoiesis

To further define the functional impact of METTL14 arginine methylation in development, we generated a METTL14 arginine methylation-deficient mouse model. Briefly, we co-injected the gRNA targeting the mouse Mettl14 gene, the donor vector containing the 13 R-to-K (RK) mutation cassette, and Cas9 mRNA into fertilized mouse eggs and subsequently screened the newborn pups for the targeted RK mutation (Fig. 6A). In addition to direct Sanger sequencing confirmation, we also developed a PCR strategy to distinguish METTL14 WT and RK mutant alleles for genotyping (Appendix Fig. S5A). Using arginine methylation-specific antibodies, we detected METTL14 ADMA modification, but not MMA or SDMA modification, in WT mouse tissues, such as spleen and thymus (Fig. 6B; Appendix Fig. S5B). Importantly, METTL14 ADMA modification was decreased in the heterozygous and completely abolished in the homozygous RK mutant mice. Consistent with our recent publication that METTL14 arginine methylation is important for cellular m⁶A homeostasis and the expression of genes involved in the Fanconi Anemia (FA) pathway (Wang et al, 2021), we observed a significant reduction of mRNA m⁶A (Appendix Fig. S5C) and the expression levels of several key genes in the FA pathway, including BRCA1, ATRIP, and PALB2, in thymus tissues from the METTL14 arginine methylation deficient RK mutant compared to their WT counterparts (Fig. 6C). Furthermore, we established mouse embryonic fibroblast (MEF) cells from littermate embryos and found that RK mutant MEFs exhibited elevated phosphorylation of histone H2AX on serine 139 (γH2AX) compared to WT MEFs (Appendix Fig. S5D), indicative of an increased genome instability, likely due to reduced DNA repair gene expression. Similarly, knockdown SMN expression also led to elevated DNA damage signals in HeLa cells (Appendix Fig. S5E).

Although we were able to obtain homozygous *Mettl14* RK mutant mice and they did not exhibit gross abnormalities during the experimental period (up to 52 weeks), the proportion of homozygous mutants (both male and female) obtained from heterozygous-to-heterozygous breeding was only about 10%, which is more than 50% reduction from the expected Mendelian ratio (Appendix Fig. S6A). Further analysis of the E12.5 embryos yielded

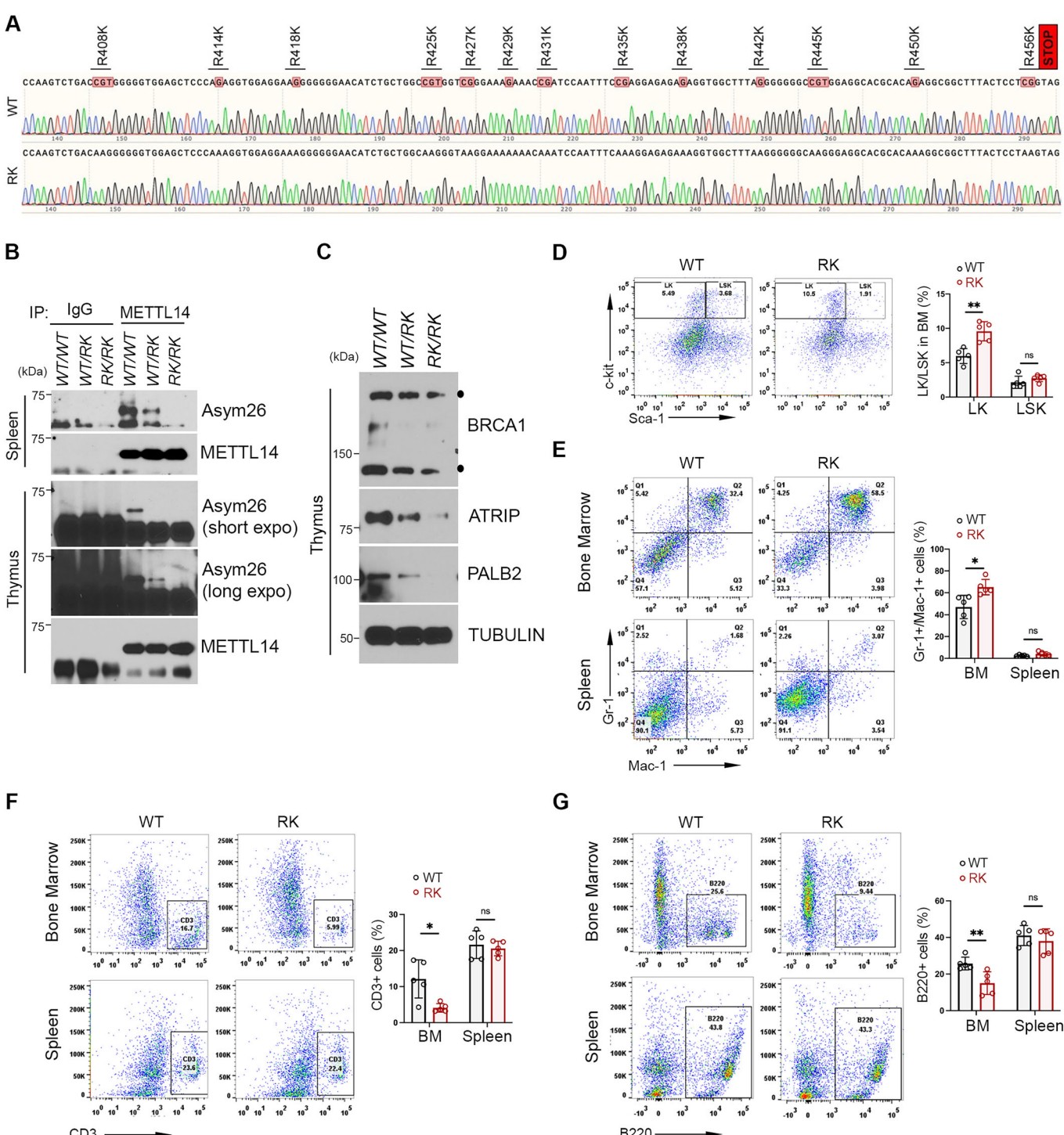

**Figure 6. METTL14 arginine methylation deficiency impairs normal hematopoiesis.**

(A) Sanger sequencing confirmation of Mettl14 arginine methylation deficient mouse model. In total, 13 arginine resides located at the C-terminal disordered region of Mettl14 were mutated to lysine residues. (B) The arginine methylation levels of Mettl14 in WT, heterozygous (WT/RK), and homozygous (RK/RK) mice were detected by IP-western blot using tissue lysates from mouse spleen and thymus. (C) The protein expression levels of several DNA repair genes, including BRCA1, ATRIP, and PALB2, were detected by western blot using Thymus tissues from WT, WT/RK, and RK/RK mice. (D) Mettl14 arginine methylation deficiency increases the population of primitive hematopoietic cells (LK cells). **$P = 0.0019$. (E) Mettl14 arginine methylation deficiency increases the population of myeloid (Mac1+Gr1+) cells in the BM, but not in the spleen. *$P = 0.013$. (F) Mettl14 arginine methylation deficiency decreases the B lymphoid (B220+) population in the BM, but not in the spleen. *$P = 0.011$. (G) Mettl14 arginine methylation deficiency decreases the T lymphoid (CD3+) population in the BM, but not in the spleen. Data from three independent replicates were analyzed by Student's $t$ test and shown as mean ± SD. **$P = 0.0097$. Source data are available online for this figure.

similar results (Appendix Fig. S6A), suggesting that Mettl14 hypomethylation impacts embryonic development in mice. The exact embryonic development defects are still under investigation, however, beyond the scope of this study.

Previous studies have reported critical roles of METTL14-mediated m6A pathway in regulating normal hematopoietic stem cells (HSCs) self-renewal, myeloid differentiation, and B cell development (Weng et al, 2018; Yao et al, 2018; Zheng et al, 2020). Thus, we investigated the impacts of METTL14 hypomethylation on normal hematopoiesis. By analyzing the complete blood counts of peripheral blood (PB) from WT and Mettl14 RK mutant mice, we did not observe any significant changes in white blood cell (WBC), lymphocyte (LYM), red blood cell (RBC), neutrophil (NEU), and monocyte (MONO) counts (Appendix Fig. S6B–E). Further, to evaluate the impact of Mettl14 hypomethylation on HSCs, we conducted flow cytometry analysis of bone marrow mononuclear cells (BM MNCs) and found a slight, but significant, increase in the population of primitive hematopoietic cells, especially the LK populations (Fig. 6D). Moreover, we evaluated the frequency of functional hematopoietic lineages, including B lymphoid (B220+), T lymphoid (CD3+), and myeloid (Mac1+Gr1+) cells in the BM of WT and Mettl14 RK mutant mice. As shown in Fig. 6E–G, Mettl14 hypomethylation significantly increased the myeloid cells, while decreasing the T and B lymphoid cells, akin to Mettl14 knockout (KO) mice (Weng et al, 2018; Yao et al, 2018; Zheng et al, 2020). Consistent with the observations in MEFs, the bone marrow from RK mutant mice showed a significantly reduced mRNA m6A and elevated genome instability marked by the γH2AX (Appendix Fig. S6F,G). To further evaluate the impact of Mettl14 hypomethylation on hematopoiesis, we performed HSC reconstitution assays by transplanting CD45.2 WT HSCs or CD45.2 Mettl14 RK HSCs into lethally irradiated CD45.1 recipient mice (Appendix Fig. S7A). In contrast to WT control, Mettl14 RK significantly impaired self-renewal capacity of CD45.2 HSCs, as evidenced by reduced engraftment (Appendix Fig. S7B–D). Further analysis revealed that Mettl14 hypomethylation significantly promoted myeloid differentiation, while suppressing B cell development in the bone marrow (Appendix Fig. S7E). This is consistent with our findings in Mettl14 RK knock-in mice. In spleen and peripheral blood, Mettl14 hypomethylation decreased B cell population while increasing T cell population (Appendix Fig. S7F,G). Notably, while loss of function in the FA pathway typically results in bone marrow failure and a cancer-prone phenotype (Nalepa and Clapp, 2018), these effects were not clearly manifested in Mettl14 RK mutant mice, suggesting that compensatory mechanisms might mitigate the impact of Mettl14 hypomethylation in these animals.

## Discussion

Our study reveals a novel arginine methylation-dependent interaction between the m6A methyltransferase METTL14 and the Tudor domain of SMN. This interaction enhances METTL14's association with RNAPII and plays a crucial role in ensuring proper m6A deposition in cells. By focusing on a group of interstrand crosslinking (ICL) DNA repair genes whose m6A levels were previously shown to be sensitive to METTL14 arginine methylation, we uncover a previously unrecognized connection between SMN deficiency, m6A homeostasis, and the dysregulation of DNA

damage gene expression. Furthermore, by establishing and characterizing a METTL14 arginine methylation-deficient mouse model, we reveal the indispensable role of METTL14 arginine methylation in mouse development.

## SMN is a multifunctional methylarginine effector involved in RNA metabolism

SMN is ubiquitously expressed and localizes to both the cytoplasm and the nucleus (Mercuri et al, 2022). It harbors highly conserved domains and mainly functions as part of a multi-protein complex essential for the assembly of small nuclear ribonucleoproteins (snRNPs) (Mercuri et al, 2022). Specifically, the Tudor domain recognizes arginine methylated protein substrates, such as Sm proteins (Brahms et al, 2001; Tripsianes et al, 2011), and the evolutionarily conserved C-terminal YG-box promotes SMN oligomerization (Lorson et al, 1998). In its canonical role, SMN interacts with SDMA-modified Sm proteins, such as SmD1, SmD3, and SmB/B', and assembles them onto snRNA during snRNP maturation (Friesen et al, 2001; Grimm et al, 2013; Meister et al, 2001), a process essential for spliceosome assembly and RNA splicing. Additionally, SMN also interacts with RNAPII (Zhao et al, 2016) and many other RNA binding proteins (RBPs) (Singh et al, 2017) to regulate transcription and other RNA processing steps, such as mRNA trafficking and translation. It is possible that the global reduction in mRNA m6A levels observed upon SMN knockdown in HeLa cells and SMA patient-derived fibroblasts (Fig. 4) may potentially be an indirect consequence of altered RNA splicing. Although it is challenging to clearly distinguish SMN's role in splicing regulation from its influence on m6A homeostasis at the whole transcriptome level, our data show that the splicing patterns of DNA repair genes, including BLM, ATRIP, and FANCM, were not affected by SMN knockdown (Appendix Fig. S3C), arguing that the SMN-mediated regulation of m6A deposition on these transcripts is unlikely a direct result of splicing alterations. Another intriguing question is how SMN facilitates the interaction between METTL14 and the CTD of RNAPII via Tudor–methylarginine binding, specifically whether a single Tudor domain can simultaneously recognize methylarginine marks on two separate substrates. Recent studies have shown that arginine methylation and Tudor domain contribute to the formation of biomolecular condensates (Binda et al, 2023; Courchaine et al, 2021). We therefore speculate that SMN's ability to multimerize or participate in condensate formation may help coordinate interactions with multiple arginine methylated substrates. Although RNA is not required for the SMN–METTL14 interaction (Appendix Fig. S1A), given the critical role of RNA in regulating condensate assembly and dynamics (Wadsworth et al, 2024), it remains worthwhile to investigate where RNA contributes to SMN-mediated METTL14–RNAPII interactions.

## SMN deficiency, m6A homeostasis, and the regulation of DNA damage gene expression

SMA is a progressive neurodegenerative disease caused by a severe deficiency of SMN protein that leads to α-lower motor neuron loss and subsequent muscle atrophy (Crawford and Pardo, 1996; Kariya et al, 2008; Lefebvre et al, 1995). Interestingly, preclinical studies in patients and mouse models have recently highlighted additional

systematic aspects of this disease, with dysfunction in heart, kidney, liver, pancreas, spleen, bone, and immune systems (DiSabato et al, 2024; Leow et al, 2024; Singh et al, 2021; Sun et al, 2016; Yeo and Darras, 2020). How loss of function of SMN1 links to dysfunction in peripheral tissues and organs remains unknown. The arginine methylation-dependent interaction between METTL14 and SMN, along with the global reduction in cellular m6A levels observed in SMN knockdown cells and in SMA patient-derived fibroblasts, uncovers a novel molecular pathway that might underline the systematic aspects of SMA pathogenesis. It is estimated that one in four mRNA transcripts are subjected to m6A modification (Boulias and Greer, 2023). Given the broad influence of m6A homeostasis on RNA processing, including splicing, export, translation, and degradation, SMN deficiency-induced m6A dysregulation could impact gene expression through mechanisms extending beyond its well-characterized role in the maturation of small nuclear ribonucleoprotein complexes (snRNPs) (Singh et al, 2017). One limitation of this study is that the extent to which SMN deficiency impacts m6A homeostasis is yet to be determined by mapping differential m6A sites using genome-wide approaches, such as m6A RNA immunoprecipitation (RIP)-seq. It is likely that the impact of SMN on m6A homeostasis is context-dependent and cell-type specific. Although ongoing efforts aim to explore this in greater depth, we focused on a specific subset of m6A sites influenced by arginine methylation of METTL14, particularly those affecting ICL DNA repair genes. Our study revealed that m6A deposition on these targets are regulated through an arginine methylation-dependent interaction between METTL14 and SMN. Consequently, SMN deficiency results in down-regulation of ICL DNA repair gene expression and sensitization to interstrand DNA crosslinking agents. These results provide a critical molecular link between SMN deficiency and genome instability, a hallmark of SMA pathology that had remained poorly understood at the molecular level (Jangi et al, 2017; Karyka et al, 2022).

## METTL14 arginine methylation and DNA damage response

PRMTs play important roles in regulating DNA damage response, largely by directly methylating proteins involved in DNA repair pathways (Brobbey et al, 2022). For example, several DNA damage repair proteins have been identified as substrates of PRMTs, including MRE11, BRCA1, and 53BP1 (Boisvert et al, 2005a; Boisvert et al, 2005b; Montenegro et al, 2020). The MEFs derived from PRMT1 knockout mice display spontaneous DNA damage and polyploidy (Yu et al, 2009). In cancer, PRMT1 expression predicts sensitivity to platinum-based chemotherapy in patients with ovarian and pancreatic cancers, and PRMT1 inhibition reduces the clonogenic growth of cancer cells exposed to low doses of cisplatin, sensitizing them to apoptosis (Giuliani et al, 2021; Ku et al, 2024; Matsubara et al, 2021; Nguyen et al, 2024). The interstrand crosslinking agents, including platinum-based agents, are still the first-line chemotherapy treatment for many solid cancers (Brown et al, 2019). Our discovery that PRMT1-catalyzed METTL14 arginine methylation plays a critical role in promoting ICL DNA repair gene expression reveals a new way of sensitizing cancer cells to ICL chemotherapy through targeting the METTL14 arginine methylation mediated m6A pathway.

Although recent development of small molecule inhibitors of PRMTs have shown promising potential as a novel therapeutic approach for treating cancers, a major limitation is that PRMTs are essential for basic cellular functions, making broad PRMT inhibition intolerable due to adverse effects on normal tissues. In this case, targeting Tudor domain-mediated SMN interaction with METTL14 provides a reasonable alternative. Indeed, several studies have reported on taking advantage of chemical inhibition of the Tudor–methylarginine interaction to specifically disrupt arginine methylation-mediated processes. For example, a bivalent inhibitor was developed to block the interaction of TDRD3 Tudor domain with the arginine methylated G3BP1 protein, the core nucleating factor of stress granule formation (Fan et al, 2024). More relevant to this study, a 4-iminopyridine scaffold, namely compound 1, was recently identified to target the Tudor domain of SMN (Fan et al, 2024). Compound 1 selectively binds the methylarginine-interacting, but not the methyllysine-interacting Tudor domain. However, it shows only marginal distinction among the highly homologous methylarginine-binding Tudor domains of SMN, SMNDC1 (also known as SPF30), and TDRD3. This is generally in line with the structural features of methylarginine-binding Tudor domain—based on their primary amino acid sequence, it is not possible to predict the binding specificity of Tudor domains (Chen et al, 2011). Recently, Wang et al reported that METTL14 arginine methylation can also be recognized by SPF30, a highly conserved homolog of SMN, though the biological function of this interaction was not clear (Wang et al, 2023a).

A key unresolved question is whether the level of METTL14 arginine methylation increases in response to DNA damage to facilitate the proper expression of DNA repair genes. Our efforts to address this question remain inconclusive, partly due to the technical challenges in detecting METTL14 arginine methylation, which relies on immunoprecipitation followed by western blot detection. Nevertheless, others have reported that Cisplatin treatment could either induce PRMT1 expression (Li et al, 2024) or promote its chromatin association (Musiani et al, 2020), both of which could potentially lead to increased METTL14 arginine methylation.

## Methods

**Reagents and tools table**

| Reagent/resource | Reference or source | Identifier or catalog number |
|---|---|---|
| **Experimental models** | | |
| HEK293T | ATCC | CRL-1573 |
| HeLa | ATCC | CCL-2 BSL 2 |
| GM08315 | A gift from Dr. Stéphane Richard | |
| GM03813 | A gift from Dr. Stéphane Richard | |
| GM09677 | A gift from Dr. Stéphane Richard | |
| Mettl14 methylation-deficient mouse model (*Mettl14RK*) | This study | |
| Mettl14 WT mouse embryonic fibroblasts (MEF) | This study | N/A |
| *Mettl14RK* mouse embryonic fibroblasts (MEF) | This study | N/A |

| Reagent/resource | Reference or source | Identifier or catalog number |
|---|---|---|
| **Recombinant DNA** | | |
| pcDNA3/Flag-METTL14 | Addgene | 53740 |
| p3xFlag-CMV7.1 | Sigma | E7533 |
| p3xFlag-CMV7.1 METTL14 (WT) | This study | N/A |
| p3xFlag-CMV7.1 METTL14 (1-400) | This study | N/A |
| p3xFlag-CMV7.1 METTL14 (5RK) | This study | N/A |
| p3xFlag-CMV7.1 METTL14 (8RK) | This study | N/A |
| p3xFlag-CMV7.1 METTL14 (10RK) | This study | N/A |
| p3xFlag-CMV7.1 METTL14 (13RK) | This study | N/A |
| p3xFlag-CMV7.1 SMN | This study | N/A |
| p3xFlag-CMV7.1 SMN E134K | This study | N/A |
| p3xFlag-CMV7.1 RNA Pol II | This study | N/A |
| p3xFlag-CMV7.1 RNA Pol II CTD | This study | N/A |
| pGEX-4T-1 | Amersham | 27458001 |
| pGEX-4T-1-METTL14 (WT) | This study | N/A |
| pGEX-4T-1-TDRD3 Tudor | This study | N/A |
| pGEX-4T-1-TDRD3 Tudor E691K | This study | N/A |
| pGEX-4T-1-SMN Tudor | This study | N/A |
| pGEX-4T-1-SMN Tudor E134K | This study | N/A |
| pGEX-4T-1-SMN Tudor A111G | This study | N/A |
| pGEX-4T-1-SMN Tudor I116F | This study | N/A |
| pGEX-4T-1-SMN Tudor Q136E | This study | N/A |
| pGEX-4T-1-PRMT1 | This study | N/A |
| pCMV-HA | Takara | 635690 |
| pCMV-HA-METTL14 | This study | N/A |
| pCMV-Myc | Takara | 635689 |
| pCMV-Myc-PRMT5 | This study | N/A |
| **Antibodies** | | |
| Rabbit anti-METTL3 antibody | Bethyl Laboratories | A301-567A |
| Rabbit anti-METTL14 antibody | Sigma | HPA038002 |
| Rabbit anti-YTHDF1 antibody | ABclonal | A13260 |
| Rabbit anti-YTHDF2 antibody | ABclonal | A15616 |
| Rabbit anti-YTHDF3 antibody | ABclonal | A8395 |
| Rabbit anti-YTHDC1 antibody | ABclonal | A7318 |
| Rabbit anti-YTHDC2 antibody | ABclonal | A15004 |
| Rabbit anti-FTO antibody | ABclonal | A1438 |
| Rabbit anti-ALKBH5 antibody | ABclonal | A11684 |
| Rabbit anti-m6A antibody | Sigma | ABE572-I |
| Rabbit anti-m6A antibody | Synaptic Systems | 202003 |
| Mouse anti-RNA Pol II antibody | Active Motif | 39097 |
| Rabbit anti-BLM antibody | Bethyl Laboratories | A300-110A |
| Rabbit anti-FANCD2 antibody | Novus Biologicals | NB100-182SS |
| Rabbit anti-BRCA1 antibody | ABclonal | A11549 |
| Rabbit anti-ATRIP antibody | ABclonal | A7139 |

| Reagent/resource | Reference or source | Identifier or catalog number |
|---|---|---|
| Rabbit anti-FANCM antibody | Proteintech | 12954-1-AP |
| Rabbit anti-PALB2 antibody | Proteintech | 14340-1-AP |
| Mouse anti-SMN antibody | Cell Signaling Technology | 12976S |
| Mouse anti-β-ACTIN antibody | Sigma | A5441 |
| Mouse anti-Tubulin antibody | Cell Signaling Technology | 2144 |
| Mouse anti-Flag antibody | Sigma | F3165 |
| Rabbit anti-HA antibody | Cell Signaling Technology | 3724S |
| Rabbit anti-PRMT1 antibody | Bethyl Laboratories | A300-722A |
| Rabbit anti-PRMT5 antibody | Cell Signaling Technology | 2252 |
| Rabbit Asymmetric Di-Methyl Arginine (ADMA)antibody | Cell Signaling Technology | 13522 |
| Rabbit anti-Symmetric Di-Methyl Arginine (SDMA) antibody | Cell Signaling Technology | 13222 |
| Rabbit anti-Mono-Methyl Arginine (MMA) antibody | Cell Signaling Technology | 8015 |
| Rabbit anti-ADMA (ASYM26) antibody | A gift from Dr. Stéphane Richard | |
| Mouse anti-γ-H2AX antibody | Sigma | 05-636 |
| Normal Rabbit IgG | Cell Signaling Technology | 2729 |
| Normal Mouse IgG | Santa Cruz Biotechnology | sc-2025 |
| Donkey Anti-Rabbit HRP Secondary Antibody | GE Healthcare | NA934V |
| Goat anti-Mouse Alexa Fluor 555 Secondary Antibody | Invitrogen | A-21422 |
| Goat Anti-Mouse HRP Secondary Antibody | Invitrogen | 62-6520 |
| Mouse anti-eFlourTM 450 antibody | eBioscience | 88-7772-72 |
| Mouse anti-c-Kit-APC antibody | eBioscience | 17-1171-82 |
| Mouse anti-Sca-1-PE antibody | eBioscience | 12-5981-82 |
| Mouse anti-CD3-APC antibody | BD Bioscience | 100236 |
| Mouse anti-B220-APC/Cyanine7 antibody | BD Bioscience | 561102 |
| Mouse anti-CD11b-Brilliant Violet 421 (Mac1) antibody | BD Bioscience | 562605 |
| Mouse anti-Gr1-PerCP-Cyanine5.5 antibody | Thermo Fisher Scientific | 45-5931-80 |
| Mouse anti-CD45.1 antibody | Thermo Fisher Scientific | 17-0453-82 |
| Mouse anti-CD45.2 antibody | BD Bioscience | 553772 |
| **Oligonucleotides and other sequence-based reagents** | | |
| Control siRNA | Dharmacon | D-001810-10 |
| PRMT1 siRNA | Qiagen | SI02663493 |
| PRMT5 siRNA | Dharmacon | L-015817-00-0005 |

| Reagent/resource | Reference or source | Identifier or catalog number |
|---|---|---|
| SMN siRNA | Dharmacon | L-011108-00-0005 |
| METTL14 siRNA | Dharmacon | L-014169-02-0005 |
| RT-qPCR primers | This study | Appendix Table S1 |
| MeRIP-qPCR primers | This study | Appendix Table S1 |
| Mettl14WT/RK mouse genotyping primers | This study | Appendix Table S1 |
| Splicing primers | This study | Appendix Table S1 |
| **Chemicals, enzymes and other reagents** | | |
| S-adenosyl-l-[methyl-$^3$H] methionine, (SAM[$^3$H]) | PerkinElmer | NET155V250UC |
| MS023 | MedChemExpress | HY-19615 |
| EPZ015938 (GSK3326595) | MedChemExpress | HY-101563 |
| EPZ015666 (GSK3235025) | Selleck Chemicals | S7748 |
| Cisplatin | APExBIO | A8321 |
| MMC | Cayman Chemical | 11435 |
| RNase A | Thermo Scientific | EN0531 |
| DAPI | Sigma | D9542 |
| STM2457 | MedChemExpress | HY-134836 |
| **Software** | | |
| FlowJo | N/A | N/A |
| ImageJ | N/A | N/A |
| **Other** | | |
| *E. coli* DH5α | New England Biolabs | C2987H |
| *E. coli* BL21 | New England Biolabs | C2530H |
| Lipofectamine™ 2000 Transfection Reagent | Invitrogen | 11668019 |
| PolyJet™ in vitro DNA transfection reagent | SignaGenlaboratories | SL100688 |
| TRIzol™ Reagent | Thermo Fisher Scientific | 15596018 |
| Pierce™ Protease Inhibitor Tablets, EDTA-free | Thermo Fisher Scientific | A32965 |
| Pierce™ Protein A/G UltraLink™ Resin | Thermo Fisher Scientific | 53133 |
| Anti-FLAG® M2 Magnetic Beads | Sigma | M8823 |
| DreamTaq PCR Master Mixes (2X) | Thermo Fisher Scientific | K1082 |
| Phusion™ High-Fidelity DNA Polymerase | New England Biolabs (NEB) | M0530 |

## Cell lines and reagents

HEK293 and HeLa cells were obtained from ATCC. SMA patient-derived fibroblasts (GM03815, GM03813, and GM09677) were kindly provided by Dr. Stephane Richard (McGill University). All cell lines were cultured in DMEM supplemented with 10% Fetal Bovine Serum (FBS) and maintained at 37 °C with 5% $CO_2$. Lipofectamine 2000 (Cat# 11668019) and Lipofectamine RNAi MAX (Cat# 13778150) were purchased from Thermo Fisher Scientific. Anti-FLAG M2 Magnetic Beads (M8823) was purchased from Sigma. The type I PRMT inhibitor MS023 (Cat# HY-19615) and PRMT5 inhibitor EPZ015938 (Cat# HY-101563) were purchased from MedChemExpress. Phusion high-fidelity DNA Polymerase (Cat# M0530L) and all restriction enzymes were purchased from New England Biolab. The site-directed mutagenesis kit (Cat# 200523) was purchased from Agilent Technologies.

## Antibodies and plasmids

The following antibodies were used for either IP or western blot analysis: anti-METTL14 (HPA038002, Sigma), anti-METTL3 (A301-567A, Bethyl Laboratories), anti-YTHDF1 (A13260, ABclonal), anti-YTHDF2 (A15616, ABclonal), anti-YTHDF3 (A8395, ABclonal), anti-YTHDC1 (A7318, ABclonal), anti-YTHDC2 (A15004, ABclonal), anti-FTO (A1438, ABclonal), anti-ALKBH5 (A11684, ABclonal), anti-m6A (ABE572-I-100UG, Sigma), anti-PRMT1 (A300-722A, Bethyl Laboratories), anti-BLM (A300-110A, Bethyl Laboratories), anti-FANCD2 (NB100-182SS, Novus Biologicals), anti-BRCA1 (A11549, ABclonal), anti-SMN (12976S, Cell Signaling Technology), anti-ATRIP (A7139, ABclonal), anti-FANCM (12954-1-AP, Proteintech), anti-PALB2 (14340-1-AP, Proteintech), anti-Flag (F3165, Sigma), anti-β-ACTIN (A5441, Sigma), anti-Tubulin (2144, Cell Signaling Technology), anti-RNAPII (39097, Active motif), rabbit anti-HA (3724S, Cell Signaling Technology), anti-SDMA (13222S, Cell Signaling Technology), and anti-ADMA (13522S, Cell Signaling Technology). The ASYM26 antibody was kindly provided by Dr. Stéphane Richard (McGill University).

Flag-METTL3 (#53739), Flag-METTL14 (#53740), Flag-WTAP (#53741), and Flag-RNA Pol II (#35175) were purchased from Addgene. Human METTL14 cDNA was cloned into pGEX-6P-1, pCMV-HA (Clontech), p3xFlag-CMV-7.1 (Sigma) vectors. All R-to-K mutants of METTL14 were generated using a site-directed mutagenesis kit (Agilent Technologies). pGEX-Tudor (SMN) and pGEX-Tudor (TDRD3) and Flag-SMN have been previously described (Yang et al, 2015).

## Recombinant protein expression and purification

GST-tagged constructs were transformed into *Escherichia coli* BL21 (DE3) cells and grown to an $OD_{600}$ of 0.6. Expression was induced by adding 1 mM Isopropyl β-D-1-thiogalactopyranoside (IPTG), and cells were cultured for 16 h at 16 °C. Cells were lysed by sonication in binding buffer (140 mM NaCl, 2.7 mM KCl, 10 mM $Na_2HPO_4$, and 1.8 mM $KH_2PO_4$, pH 7.4) and centrifuged at $21,000 \times g$ for 15 min. The supernatant was incubated with Glutathione Sepharose 4B resin (17-0756-01; Cytiva) overnight at 4 °C. The GST-tagged proteins were eluted with 10 mg/ml reduced L-Glutathione in elution buffer (100 mM Tris-HCl, pH 7.4, with 150 mM NaCl) after washing three times with PBS buffer.

## RNA interference

Small interfering RNA (siRNA) targeting human SMN1 (Cat# L-011108-00-0005) and the control siRNA (Cat# D-001810-10) were

purchased from Dharmacon. The siRNA targeting human PRMT1 (Cat# 1027417) was purchased from Qiagen. For all the knockdown experiments, cells were transfected with siRNA at a final concentration of 10 nM using Lipofectamine RNAi MAX for 72 h. The knockdown efficiency was confirmed by either RT-qPCR or western blot detection.

## Reverse transcription quantitative PCR (RT-qPCR)

Total RNA was extracted using TRIzol reagent (15596-018; Thermo Fisher Scientific). Reverse transcription was performed using a High-Capacity cDNA Reverse Transcription Kit (4368814; Thermo Fisher Scientific). In total, 2 μl of tenfold diluted cDNA was used for qPCR analysis using Power SYBR Green PCR Master Mix (43-687-06; Thermo Fisher Scientific). Real-time qPCR was performed on a CFX96 Real-time System C1000 Touch Thermal Cycler (Bio-Rad Laboratories), according to the manufacturer's instructions. The comparative cycle threshold ($C_T$) method ($\Delta\Delta C_T$) was used to quantify relative changes in gene expression. $C_T$ values were normalized by subtracting β-actin $C_T$ values from target gene $C_T$ values for each sample. All amplifications were done in triplicate. Data analysis was performed using the Bio-Rad CFX Manager 3.1. The primer sequences are listed in Appendix Table S1.

## Co-immunoprecipitation (co-IP)

Cells from a 10-cm plate were washed with 1X PBS and lysed with 1 ml of co-IP buffer (50 mM Tris-HCl [pH 7.4], 150 mM NaCl, 15 mM MgCl$_2$, 5 mM EDTA, and 0.1% Nonidet P-40), containing a cocktail of protease inhibitors (A32965; Thermo Fisher Scientific). Cell extracts were briefly sonicated and centrifuged at $21,000 \times g$ for 10 min to remove insoluble debris. Cell lysates were incubated with antibodies overnight at 4 °C, followed by incubation with Protein A/G polyacrylamide beads (53133; Thermo Fisher Scientific) for 2 h. Beads were then washed three times with co-IP buffer, and bound proteins were eluted by SDS-PAGE loading buffer. Proteins were resolved on an SDS-PAGE gel and analyzed by western blot analysis.

## GST pull-down

Cells were lysed in lysis buffer containing 20 mM Tris-HCl (pH 7.4), 150 mM NaCl, 0.1% NP-40, and protease inhibitors. After removing insoluble debris, the cell lysates were incubated with purified GST-tagged recombinant proteins with gentle rocking overnight at 4 °C. Glutathione Sepharose beads were added to the protein and lysate mixture and incubated with gentle rocking at 4 °C for 2 h. The mixture was centrifuged, the supernatant was discarded, and the beads were washed three times with the cell lysis buffer. After centrifuging again, the pellet was eluted in 30 μl 2× SDS sample buffer. The samples were loaded on SDS-PAGE gels and analyzed by western blot using the indicated antibodies.

## Immunofluorescence assay

The immunofluorescence assay was performed as previously described (Huang et al, 2018; Narayanan et al, 2017; Shen et al, 2024). In brief, either HeLa cells or MEFs were seeded onto coverslips and cultured in a 24-well plate. The HeLa cells were transfected with siRNA targeting SMN for 72 h before they were fixed with 4% formaldehyde in PBS for 15 min at room temperature (handling was done in a fume hood due to paraformaldehyde's toxicity). Following fixation, cells were rinsed twice in PBS for 5 min each and permeabilized with 0.3% Triton X-100 in PBS for 10 min at room temperature. This was followed by two additional PBS washes, each for 5 min. Blocking was performed using 3% BSA at room temperature for 1 h. Cells were then incubated with the indicated primary antibody (diluted in 3% BSA) overnight at 4 °C. On the following day, cells were rinsed three times with PBS for 5 min, followed by additional incubation with a fluorescence-conjugated secondary antibody for 1 h at room temperature in the dark. After incubation, cells were washed twice with PBS for 5 min each, then incubated with 0.5 μg/ml of DAPI staining buffer for 5–10 min at room temperature. Finally, the cells were mounted to the coverslips using the ProLong Gold Antifade Mountant solution (Cat# P36930, Thermo Fisher). Images were captured using a Cytation 5 Cell Imaging Multimode Reader (Agilent).

## m⁶A dot blot

Cellular mRNA was purified using the Magnetic mRNA Isolation Kit (Cat # S1550S, New England Biolabs). Subsequently, mRNA samples were incubated in RNA incubation solution (657 μl formamide, 210 μl 37% formaldehyde solution, and 133 μl 10X MOPS) at 65 °C for 5 min, followed by immediate cooling on ice. Series diluted mRNA samples in equal volumes were loaded onto a pre-wet Nylon membrane using the Bio-Dot Apparatus (Cat# 1706545, Bio-Rad Laboratories). The membrane was subjected to UV crosslinking using the Stratalinker 2400 crosslinker (Stratagene). The membrane was subsequently stained with methylene blue (0.02% methylene blue in 0.3 M sodium acetate, pH 5.2). For m⁶A blotting. UV crosslinked membrane was washed twice with TBST, blocked with 5% nonfat dry milk for 1 h at room temperature, and incubated with an m⁶A antibody (ABE572-I-100UG, Sigma, 1:200) overnight at 4 °C. The membrane was then incubated with anti-rabbit HRP (sc-2030, Santa Cruz Biotechnology) and washed 3 times in PBST before development using the ECL HRP substrate (Cat# 34580, Thermo Fisher Scientific).

## mRNA m⁶A quantification by LC-MS/MS

Total RNA from siControl, siSMN HeLa cells, as well as SMA patient-derived fibroblasts GM03815 + /−, GM03813−/−, and GM03877−/− was isolated using TRIzol reagent (Invitrogen). The polyadenylated RNA from these cells was isolated using two rounds of purification on oligo d(T)25 magnetic beads (Thermo Fisher). In total, 25 ng of poly(A) + RNA was digested using nuclease P1 (1 U, Sigma) in 20 μl of buffer containing 20 mM NH$_4$OAc (pH 5.5) at 42 °C for 2 h, followed by the addition of FastAP buffer (2.3 μl) and alkaline phosphatase (1 U, Thermo Fisher) and incubation at 37 °C for 4 h. The sample was then filtered (0.22 μm pore size, 4 mm diameter, Millipore), and 5 μl of the solution was injected into a SCIEX Triple Quad 6500 + LC-MS/MS system. The nucleosides were separated by reverse-phase ultra-performance liquid chromatography on a C18 column (Agilent) with online mass spectrometry detection performed in positive electrospray ionization mode. The nucleosides were quantified using the nucleoside-to-base ion mass transitions of 282 to 150 (m⁶A) and 268 to 136 (A). Nucleoside concentrations were

determined by comparison to a standard curve obtained from pure nucleoside standards run with the same batch of samples. The m⁶A/A ratio was calculated based on the calibrated concentrations.

## m⁶A RNA immunoprecipitation (MeRIP)-qPCR

The m⁶A RIP-qPCR was performed as previously described (Guo et al, 2024; Wang et al, 2021), with a few modifications. In brief, cells were treated with 1% formaldehyde for 10 min for crosslinking, and were quenched by adding glycine to a final concentration of 0.25 M for 5 min. After washing with PBS, the cells were lysed in RIP buffer (50 mM Tris-HCl [pH 7.5], 150 mM NaCl, 1% NP-40, 0.5% sodium deoxycholate) supplemented with Protease Inhibitor (Cat# A32965, Thermo Fisher), PMSF (Cat# 36978, Thermo Fisher), and RNase Inhibitor (Cat# M0314L, New England Biolabs). Lysis was further sonicated using a Bioruptor Pico sonication device (Diagenode) for 10 cycles of 30 s on and 30 s off. The resulting lysates were clarified by centrifugation, and the supernatant was collected. Two microliters of m⁶A antibody was added, and the samples were incubated overnight at 4 °C with gentle shaking. Next day, Dynabeads (Cat# 10002D, Thermo Fisher) were added to the mixture, followed by a 4 h incubation. Unbounded materials were removed through three washes with RIP buffer. The beads were then incubated with 100 μl of elution buffer (100 mM Tris-HCl [pH 8.0], 200 mM NaCl, 10 mM EDTA, 1% SDS) containing 0.2 mg/ml Proteinase K at 42 °C for 1 h. RNA extraction was performed by transferring 100 μl of the eluate to a new RNase-free tube, followed by adding 150 μl RNase-free water and 250 μl Phenol:Chloroform:Isoamyl Alcohol (Cat# P1944, Sigma). The mixture was vortexed thoroughly and centrifuged at 4 °C at 12000 rpm for 15 min. 200 μl of supernatant was carefully collected and mixed with 600 μl of Isopropanol/Ethanol (1:1, v/v), 20 μl of 3 M NaAc (pH 5.2) and 1 μl of GlycoBlue Coprecipitant (Cat# AM9515, Thermo Fisher). The mixture was kept at −80 °C for 1 h or at −20 °C overnight. The sample was centrifuged at 4 °C at 12,000 × g for 30 min. The resulting pellet was washed with 1 ml of fresh, 75% ice-cold ethanol. After brief centrifugation, ethanol was decanted, and any residual ethanol was aspirated to allow for a quick air-dry step. RNase-free water was added to dissolve the RNA pellet. RNA was reverse-transcribed into cDNA using the High-Capacity cDNA Reverse Transcription Kit (Cat# 4368813, Thermo Fisher) following the manufacturer's instructions. The resulting cDNA, diluted approximately tenfold, was used as a template for qRT-PCR analysis.

## Cell viability assay

To assess cell viability, cells were cultured in 96-well plates and counted at the indicated times using a CCK-8 Cell Counting Kit-8 (Dojindo), measuring the absorbance on a Cytatin 5 (Agilent) microplate reader.

## Mettl14 arginine methylation deficient (Mettl14RK) mouse model

The Mettl14 arginine methylation-deficient mouse model was generated by CRISPR/Cas-mediated genome engineering (Cyagen). The mouse Mettl14 gene (NCBI Reference Sequence: NM_201638.2) is located on mouse chromosome 3. To generate arginine methylation deficient Mettl14, all the methylated arginine residues were mutated to lysine residues to abolish methylation,

while retain positive charge. Briefly, the pR408K(CGT to AAG), pR414K(AGA to AAA), pR418K(AGG to AAG), pR425K(CGT to AAG), pR427K(CGG to AAG), pR429K(AGA to AAA), pR431K(CGA to AAA), pR435K(CGA to AAA), pR438K(AGA to AAA), pR442K(AGG to AAG), pR445K(CGT to AAG), pR450K(AGA to AAA), pR456K(CGG to AAG) were introduced into exon 11. To engineer the targeting vector, homology arms was generated by PCR using Bacterial Artificial Chromosome (BAC) clone RP24-338F17 as template. Cas9 and guide RNA (gRNA) was co-injected into fertilized eggs with targeting vector for mice production. F0 founder animals were genotyped by PCR followed by Sanger sequencing analysis. Subsequently, positive F0 founders were bred to wild-type (WT) mice to test germline transmission and F1 animal generation. gRNA sequences are: gRNA1 TAGA-TATGTACCCCAGGCGT and gRNA2 GAAGTGTGCATTAGGA ATAC.

For mouse genotyping, mouse tails were collected for DNA extraction by incubating with DirectPCR buffer (Cat# 102-T, VIAGEN) supplemented with proteinase K at 65 °C overnight followed by boiling for 20 min. Subsequent genotyping was performed using PCR. WT forward: CCCGCTTTATTTCAGGC TGGCTC; reverse: GCCTCTGTGCGTGCCTCCACGG; RK forward: CCCGCTTTATTTCAGGCTGGCTC (same as the WT forward primer); reverse: GCCTTTGTGCGTGCCTCCCTTG.

## Flow cytometric analysis

The assessment of Mettl14 hypomethylation mutation on normal hematopoiesis was conducted as previously described, with some modifications (Gu et al, 2021; Han et al, 2023). Both male and female wild-type (WT) and Mettl14RK knock-in mice were included in the experiments. Peripheral blood (PB) was collected from the tail vein, and complete blood counts were analyzed using an element HT5 (HESKA) instrument following the manufacturer's protocol. Bone marrow (BM) and spleen (SP) mononuclear cells were collected and lysed with Ammonium Chloride Solution (07850, STEMCELL Technologies) to remove red blood cells. The cells were then incubated with antibodies in FACS buffer (00-4222-26, eBioscience) on ice for 30 min in the dark, followed by analysis via flow cytometry. Hematopoietic progenitors in BM, including the linage (Lin)⁻ Sca-1⁺ c-Kit⁺ (LSK) compartment and the linage (Lin)⁻ Sca-1⁻ c-Kit⁺ (LK) compartment, were analyzed. Functional hematopoietic lineages in BM and SP including B lymphoid (B220⁺), T lymphoid (CD3⁺), and myeloid (Mac1⁺ Gr1⁺) cells were also analyzed. The following antibodies were used for flow cytometry: eFlourTM 450 (88-7772-72, eBioscience), c-Kit-APC (17-1171-82, eBioscience), Sca-1-PE (12-5981-82, eBioscience), CD3-APC (100236, BD Bioscience), B220-APC/Cyanine7 (561102, BD Bioscience), CD11b-Brilliant Violet 421 (Mac1, 562605, BD Bioscience) and Gr1-PerCP-Cyanine5.5 (45-5931-80, Thermo Fisher Scientific).

## Hematopoietic stem cells (HSCs) reconstitution

Lineage-negative (Lin-) HSCs were enriched from 6- to 8-week-old CD45.2 wild-type (WT) and Mettl14 RK knock-in mice using the Lineage Cell Depletion Kit (130-090-858, Miltenyi Biotec). A total of $1 \times 10^6$ enriched Lin⁻ HSCs from either WT or Mettl14 RK knock-in mice were transplanted into lethally irradiated 6- to 8-

week-old NCI C57BL/6.SJL (CD45.1$^+$) recipient mice ($n = 3$ per group). Six weeks post-transplantation, recipient mice were euthanized, and peripheral blood (PB), spleen (SP), and bone marrow (BM) samples were collected for flow cytometric analysis. The following antibodies were used: CD45.1 (17-0453-82, Thermo Fisher Scientific), CD45.2 (553772, BD Bioscience), CD3-APC (100236, BD Bioscience), B220-APC/Cyanine7 (561102, BD Bioscience), CD11b-Brilliant Violet 421 (Mac1, 562605, BD Bioscience) and Gr1-PerCP-Cyanine5.5 (45-5931-80, Thermo Fisher Scientific).

## Statistical analysis

All experiments were performed at least three times. Replicate data were presented as mean ± SD. To assess differences among groups, an unpaired two-tailed $t$ test or one-way ANOVA was utilized. Statistical significance was determined at a $P$ value of less than 0.05. Quantification of immunoblotting images was performed using ImageJ software.

# Data availability

This study includes no data deposited in external repositories.

The source data of this paper are collected in the following database record: biostudies:S-SCDT-10_1038-S44319-025-00590-7.

# Peer review information

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

## Acknowledgements

Research reported in this publication was supported by the National Institute of Health under award number R01GM133850, R35GM156316, and R01NS132083 (YY), RM1HG008935 (CH). We thank Dr. Stéphane Richard (McGill University) for sharing SMA patient-derived fibroblasts. The content is solely the responsibility of the authors and does not necessarily represent the official views of the National Institutes of Health. We thank Erin Keebaugh for providing helpful comments on scientific writing.

## Author contributions

**Yi Zhang**: Conceptualization; Data curation; Formal analysis; Validation; Investigation; Methodology; Writing—original draft. **Lei Shen**: Data curation; Formal analysis; Validation; Investigation; Methodology. **Lili Ren**: Conceptualization; Data curation; Validation; Investigation; Methodology. **Jiangbo Wei**: Data curation; Investigation; Methodology. **Hoang Quoc Hai Pham**: Data curation; Investigation; Methodology. **Xiaoqun Tao**: Conceptualization; Data curation; Formal analysis; Validation; Investigation; Methodology. **Jiamin Guo**: Data curation; Investigation; Methodology. **Zhihao Wang**: Conceptualization; Data curation; Investigation; Methodology. **Binghui Shen**: Resources; Funding acquisition. **Rui Su**: Conceptualization; Data curation; Supervision; Validation; Investigation; Methodology; Writing—original draft; Writing—review and editing. **Chuan He**: Conceptualization; Supervision. **Yanzhong Yang**: Conceptualization; Resources; Supervision; Funding acquisition; Methodology; Writing—original draft; Project administration; Writing—review and editing.

Source data underlying figure panels in this paper may have individual authorship assigned. Where available, figure panel/source data authorship is listed in the following database record: biostudies:S-SCDT-10_1038-S44319-025-00590-7.

## Disclosure and competing interests statement

CH is a scientific founder, a member of the scientific advisory board and equity holder of Aferna Bio and Ellis Bio, a scientific cofounder and equity holder of Accent Therapeutics, and a member of the scientific advisory board of Rona Therapeutics and Element Biosciences. The remaining authors declare no competing interests.

