## [Peer Review File · EMBO Reports]

Arginine methylation-dependent METTL14-SMN interaction regulates RNA m6A homeostasis

Yanzhong Yang, Yi Zhang, Lei Shen, Lili Ren, Jiangbo Wei, Hoang Quoc Hai Pham, Xiaoqun Tao, Jiamin Guo, Zhihao Wang, Binghui Shen, Rui Su, and Chuan He

Corresponding author(s): Yanzhong Yang (yyang@coh.org)

Review Timeline:

Submission Date:	7th Feb 25
Editorial Decision:	27th Mar 25
Revision Received:	31st Jul 25
Editorial Decision:	12th Sep 25
Revision Received:	15th Sep 25
Accepted:	23rd Sep 25

Editor: Esther Schnapp

Transaction Report:

Dear Dr. Yang,

Thank you for your patience while your manuscript was peer-reviewed at EMBO reports. We have now received the full set of referee reports as well as referee cross-comments that are pasted below.

As you will see, the referees acknowledge that the findings are potentially interesting. However, referee 1 raises several concerns and does not support the publication of this study by EMBO reports. I asked referee 2 and 3 for cross-comments and both referees - while agreeing with referee 1 on several points - concur that you should be given a chance to revise your study. I would therefore like to invite you to do so. All concerns should be addressed, except for the HSC reconstitution in lethally irradiated mice, which is not a strict requirement but certainly a good suggestion. I also think that the mouse data should stay in the ms. Please let me know in case you have any questions or comments and we can also discuss the revision requirements in a video chat, if you wish.

I would thus like to invite you to revise your manuscript with the understanding that the referee concerns must be fully addressed and their suggestions taken on board. Please address all referee concerns in a complete point-by-point response. Acceptance of the manuscript will depend on a positive outcome of a second round of review. It is EMBO reports policy to allow a single round of major revision only and acceptance or rejection of the manuscript will therefore depend on the completeness of your responses included in the next, final version of the manuscript.

We realize that it is difficult to revise to a specific deadline. In the interest of protecting the conceptual advance provided by the work, we recommend a revision within 3 months (27th Jun 2025). Please discuss the revision progress ahead of this time with the editor if you require more time to complete the revisions.

- 1) A data availability section providing access to data deposited in public databases is missing. If you have not deposited any data, please add a sentence to the data availability section that explains that.
- 2) Your manuscript contains statistics and error bars based on $n=2$. Please use scatter blots in these cases. No statistics should be calculated if $n=2$.

5) a complete author checklist, which you can download from our author guidelines <https://www.embopress.org/page/journal/14693178/authorguide>. Please insert information in the checklist that is also reflected in the manuscript. The completed author checklist will also be part of the RPF.

6) Please note that all corresponding authors are required to supply an ORCID ID for their name upon submission of a revised manuscript (<https://orcid.org/>). Please find instructions on how to link your ORCID ID to your account in our manuscript

tracking system in our Author guidelines

<<https://www.embopress.org/page/journal/14693178/authorguide#authorshippinguidelines>>

10) Regarding data quantification (see Figure Legends:

<https://www.embopress.org/page/journal/14693178/authorguide#figureformat>)

12) All Materials and Methods need to be described in the main text using our 'Structured Methods' format, which is required for all research articles. According to this format, the Methods section includes a Reagents and Tools Table (listing key reagents, experimental models, software and relevant equipment and including their sources and relevant identifiers) followed by a Methods and Protocols section describing the methods using a step-by-step protocol format. The aim is to facilitate adoption of the methodologies across labs. More information on how to adhere to this format as well as a downloadable template (.docx) for the Reagents and Tools Table can be found in our author guidelines:

An example of a Method paper with Structured Methods can be found here: <https://www.embopress.org/doi/full/10.1038/s44320-024-00037-6#sec-4>

As part of the EMBO publication's Transparent Editorial Process, EMBO reports publishes online a Review Process File (RPF)

to accompany accepted manuscripts. This File will be published in conjunction with your paper and will include the referee reports, your point-by-point response and all pertinent correspondence relating to the manuscript.

I look forward to seeing a revised form of your manuscript when it is ready.

Yours sincerely,

Referee #1:

Zhang et al.

Arginine methylation-dependent METTL14-SMN interaction regulates RNA m6A homeostasis

In this manuscript, the authors have investigated functional interactions between the Tudor domain of SMN and the methylated RGG repeats in the C-terminal IDR of METTL14. The authors have shown before that METTL14 is modified mainly to asymmetrical DMA and this modification is important for m6A methylation mediated by the METTL3-METTL14 complex. Here, they now investigate the Tudor domain proteins SMN and TDRD3 and find that SMN interacts with methylated METTL14 in a number of different pull-down experiments. Knock down of PRMT1 and inhibiting sDMA and aDMA modification by small molecules inhibited interactions between SMN-Tudor and METTL14. SMN has been linked to pol II transcription and the authors present data suggesting that SMN facilitates the interaction between METTL14 and the CTD of pol II. Knock down of SMN as well as patient-derived cell lines show reduced global m6A levels. This can be specifically observed for DNA repair genes, which have been associated to METTL14 DMA by the authors before. Finally, a mouse model lacking arginines that can be modified shows a hematopoietic phenotype but without an apparent link to SMN function.

This is a well-written manuscript and the results are presented clearly. The role of the RGG repeats in the C-terminal IDR of METTL14 is mechanistically unclear and thus the presented study is certainly relevant. Although the individual results are solid, interpretations and conclusions appear rather immature and sometimes not fully supported by the data. More specific points are listed below.

1. The link to the Tudor domain of SMN is unclear and puzzling. SMN interacts with sDMA in a very specific subset of many target proteins. There is also a clear link between PRMT5 and SMN. RGG repeats are frequently found in RBPs and many of them are indeed asymmetrically methylated, which often affects RNA binding efficiency. Here the authors analyze PRMT1, which is not much linked to SMN but do not really systematically investigate the known role of PRMT5 and sDMA for SMN function.
2. A second major issue is that SMN is involved in splicing. So, knock downs or the use of patient-derived mutants may affect splicing and thus also the m6A landscape. Thus, most data on m6A levels in this manuscript could be indirectly caused by altering the splicing process.
3. The RGG repeats in METTL14 have been shown to affect RNA substrate binding in a number of publications before. This is largely ignored and at least some of the observed interactions could be due to a loss of RNA interaction of METTL14 and thus interactions could be indirect and mediated through RNA.
4. Figure 1: SMN is nuclear and cytoplasmic. Is there a colocalization of the two proteins in IF? Is SMN binding in the context of its large SMN complex or rather as individual protein? In Figure 1B, METTL3 does not bind to SMN. However, at least a fraction of the tagged protein might form a complex with endogenous METTL14 and thus a complete absence of interaction might be rather unexpected.
5. Figure 2B: binding is already fully abolished when 5 arginines are mutated. Are the other 8 not methylated in this mutant?
6. Figure 2D: the authors mention that scavenger methylation may occur, when PRMT1 is knocked down, which could explain the interaction with SMN since PRMT5 could now be recruited and generate sDMA. One would expect an increase in SMN binding in such a model. However, a decrease is observed. Similarly, in 2G, a clear effect on interaction is only observed when

sDMA and sDMA generation is inhibited, which is not consistent with the strong effect that PRMT1 knock down has. This remains unclear and is unfortunately not followed up further.

7. Figure 3 is not convincing. The authors arrive at a model in which SMN mediates the interaction of METTL14 with the CTD of pol II. SMN interacts with the methylated CTD and also interacts with methylated DMA on METTL14. The Todor domain could only interact with the one or the other at a time? So how would such a model work? Is there any RNA-dependence in these assays?

8. Figure 4: as mentioned, the rather mild effects that are observed could be due to splicing deficits caused by compromising SMN function and may not be linked to methylation of METTL14. The mouse that has been generated does also not provide further evidence that there is a functional connection.

Referee #2:

In this manuscript by Zhang et al the authors built on their previous discovery regarding the function of methylated arginines in METTL14. Their previously shown that this methylation is important for m6A deposition, in particular on transcripts involved in DNA repair. Here they found that SMN can interact with METTL14 through its arginine methylation domain, and consequently this stimulates the function of METTL14. Lack of SMN leads to reduced m6A levels in HeLa cells, likely due to the decreased association with RNA Pol II. Interestingly SMN patient derived fibroblasts also display reduction of m6A. Lastly they engineered a METTL14 arginine methylation deficient mouse model, showing some impact on embryonic development, alteration of hematopoietic lineage and increased genome instability.

This study uncovers a novel interaction of METTL14 with SMN, which plays a critical role in spinal muscular atrophy. This provides another mechanism by which SMN alteration might contribute to the disease. The manuscript is well written and the experimental data are solid and properly interpreted.

I have only some minor comments that could help further improving the manuscript.

Figure 4:

- Panel B is not very informative. The decrease m6A level is unclear in the SMN KD
- Panel D should be quantified.

Figure 5

- Panel D: A similar experiment should be repeated with a METTL3 depletion (KD or inhibitor). Otherwise it remains unclear that the effect of SMN depletion on FANCD2 level depends on its m6A stimulating activity
- Panel G: Given the data presented in panels D-F it would be logical to also look at FANCD2 m6A and protein levels in the patients derived cells.

Figure 6

- Does the level of m6A altered in the RK/RK animals as it would be expected?

Referee #3:

The manuscript by Zhang et al explores a new link between arginine methylation and RNA m6A homeostasis. The paper is well written, experiments are carefully controlled, and overall, I am very positive to see it published in EMBORep.

Some suggestions for improvements.

Major.

Fig.2D/E, have the authors tried the same experiment with siPRMT5?

Fig.5. Given the published link between PRMT5 and DDR genes, have the authors tried to perform the experiment in Fig.5A and Fig.5C in the presence of EPZ, but more importantly of EPZ+MS023?

Fig.6B. Have the author tried to quantify MMA or SDMA? It would be important to understand if in vivo those sites are mainly ADMA or else.

Fig.6. Was the increase in gammaH2AX also observed in hematopoietic cells?

Fig.6. Given the lack of phenotype at steady state, but potential defects in HSCs, have the authors tried a HSC reconstitution in lethally irradiated mice? The stress imposed to HSCs might uncover additional phenotypes.

Minor.

Please fix some spelling mistakes, and some other minor text issues such as:

1. "knockdown of PRMT1 expression using siRNA significantly reduced the interaction of METT14 with SMN as revealed by both co-IP and GST pull down assays (Figure 2D and 2E)." METT14, instead of METTL14,
2. "consistent with the general observation of PRMT substrate scavenging (Refs)," Actually add the references.
3. Were the pulldowns described in Fig.1/2 performed in the presence of RNase?

Cross-comments from referee 2:

I agree that the comments of reviewer 1 are important, especially the link with PRMT5 should be better studied (reviewer 3 also explicitly asked for this).

Point 2 will be difficult to address.

I think all the other points could be addressed or discussed.

If the authors can revise the manuscript it could be a great story for the field so I would recommend major revisions instead of a rejection.

Cross-comments from referee 3:

After reading all the comments, I would definitely give a chance to the author to prepare a rebuttal. In the meantime, here are my thoughts.

-The link to the Tudor domain of SMN is unclear and puzzling. SMN interacts with sDMA in a very specific subset of many target proteins. There is also a clear link between PRMT5 and SMN. RGG repeats are frequently found in RBPs and many of them are indeed asymmetrically methylated, which often affects RNA binding efficiency. Here the authors analyze PRMT1, which is not much linked to SMN but do not really systematically investigate the known role of PRMT5 and sDMA for SMN function.

I agree with that, and indeed, have asked to explore more the role of PRMT5.

I disagree however on the statement that SMN only interacts with sDMA. It has been shown to also interact with ADMA in multiple papers (see as an example PMID: 22101937)

-A second major issue is that SMN is involved in splicing. So, knock downs or the use of patient-derived mutants may affect splicing and thus also the m6A landscape. Thus, most data on m6A levels in this manuscript could be indirectly caused by altering the splicing process.

Possible, and should be discussed, but I still think the mechanistic link provided by this paper is interesting. Often in biology (and this is very true for PRMT biology) there are broad effects that are hard to pick apart.

The other comments are addressable, and I think the paper still stands even if the mouse data (which indeed is not fully linked to the rest of the paper) is removed.

Response to editorial decision letter

The referees acknowledge that the findings are potentially interesting. However, referee 1 raises several concerns and does not support the publication of this study by EMBO reports. All concerns should be addressed, except for the HSC reconstitution in lethally irradiated mice, which is not a strict requirement but certainly a good suggestion. I also think that the mouse data should stay in the ms. Please let me know in case you have any questions or comments, and we can also discuss the revision requirements in a video chat, if you wish.

I would thus like to invite you to revise your manuscript with the understanding that the referee concerns must be fully addressed and their suggestions taken on board. Please address all referee concerns in a complete point-by-point response. Acceptance of the manuscript will depend on a positive outcome of a second round of review. It is EMBO reports policy to allow a single round of major revision only and acceptance or rejection of the manuscript will therefore depend on the completeness of your responses included in the next, final version of the manuscript.

We thank the reviewers for their overall positive comments on our manuscript. We agree with reviewer's critiques and have addressed them in the revised manuscript. Specifically, we now added new results showing that 1) although arginine methylation promotes METTL14 protein–protein and protein–RNA interaction (as we previously reported), the interaction between METTL14 and SMN is not dependent on RNA (new **Appendix Fig. S1A**); 2) compared to PRMT, PRMT5 weakly methylates METTL14 in vitro (new **Appendix Fig. S2A**), and PRMT5 knockdown does not significantly affect METTL14-SMN interaction (new **Appendix Fig. S2B**), suggesting that the contribution of PRMT5-catalyzed SDMA modification of METTL14 is likely only evident when type I PRMT activity is inhibited; 3) we provide additional evidence (in this response letter) to show that SMN Tudor binds to both ADMA and SDMA modified protein substrates, and that its ability to multimerize may explain its role in mediating METTL14 and RNAPII interaction. We also discussed this possibility in the revised manuscript; 4) although it is possible that other molecular pathways, such as RNA splicing, could be affected by SMN loss, which may indirectly contribute to the globally reduced m⁶A levels, we showed that the splicing of DNA repair genes that are regulated by METTL14 arginine methylation were not significantly altered by SMN knockdown (new **Appendix Fig. S3C**); 5) we further evaluated the impact of METTL14 arginine methylation deficiency on hematopoiesis by performing hematopoietic stem cell reconstitution in lethally irradiated mice (new **Appendix Fig. S7**). The results further confirmed the critical role of METTL14 methylation in normal hematopoiesis.

We have thoroughly considered all the minor comments and made changes as suggested.

We would like to thank all the reviewers for their insightful comments. Below is a detailed point-by-point response to the comments with our responses in blue text and the original comments in black. The sections of the main text that have been modified are highlighted in red.

Detailed Responses to Reviewer Comments

Referee #1:

In this manuscript, the authors have investigated functional interactions between the Tudor domain of SMN and the methylated RGG repeats in the C-terminal IDR of METTL14. The authors have shown before that METTL14 is modified mainly to asymmetrical DMA and this modification is important for m6A methylation mediated by the METTL3-METTL14 complex. Here, they now investigate the Tudor domain proteins SMN and TDRD3 and find that SMN interacts with methylated METTL14 in a number of different pull-down experiments. Knock down of PRMT1 and inhibiting sDMA and aDMA modification by small molecules inhibited interactions between SMN-Tudor and METTL14. SMN has been linked to pol II transcription, and the authors present data suggesting that SMN facilitates the interaction between METTL14 and the CTD of pol II. Knock down of SMN as well as patient-derived cell lines show reduced global m6A levels. This can be specifically observed for DNA repair genes, which have been associated to METTL14 DMA by the authors before. Finally, a mouse model lacking arginines that can be modified shows a hematopoietic phenotype but without an apparent link to SMN function.

This is a well-written manuscript, and the results are presented clearly. The role of the RGG repeats in the C-terminal IDR of METTL14 is mechanistically unclear and thus the presented study is certainly relevant. Although the individual results are solid, interpretations and conclusions appear rather immature and sometimes not fully supported by the data. More specific points are listed below.

We thank the reviewer for the highly positive comments about our work. The reviewer brought up a few important points for clarification to strengthen the manuscript. We have addressed these critiques in the revised manuscript, as detailed below.

1. The link to the Tudor domain of SMN is unclear and puzzling. SMN interacts with sDMA in a very specific subset of many target proteins. There is also a clear link between PRMT5 and SMN. RGG repeats are frequently found in RBPs and many of them are indeed asymmetrically methylated, which often affects RNA binding efficiency. Here the authors analyze PRMT1, which is not much linked to SMN but do not really systematically investigate the known role of PRMT5 and sDMA for SMN function.

We appreciate the reviewer's concern and recognize that there might be a general conception that the Tudor domain of SMN preferentially interacts with SDMA modified substrates. However, as reviewer 3 also noted (PMID: 22101937), methylarginine binding Tudor domains, such as those in SMN, TDRD3, and SPF30, do not display a strong general preference for ADMA or SDMA. Instead, their binding preference depends on the specific substrate context. For example, Tudor domain of TDRD3 prefers to interact with ADMA modified histone tails, such as H4R3me2a, over H4R3me2s. However, TDRD3 Tudor domain also interact with SDMA modified substrates (**Response Figure 1**). To further elaborate on this, we perform a series of GST pull-

down using wild type and methylarginine binding deficient Tudor domains of TDRD3 and SMN. When detected with the pan-methylarginine antibodies, the Tudor domains of both TDRD3 and SMN can pull down methylated proteins with all three types of modifications (**Response Figure 1**). Notably, there are also clear differences in their binding patterns, indicating their substrate

Figure for referee with unpublished data and its description has been removed upon request by the authors.

selectivity.

To address the role of PRMT5 in METTL14 arginine methylation and interaction with SMN, we added new data showing that compared to PRMT1, PRMT5 only weakly methylates METTL14 in vitro, and that knockdown PRMT5 does not affect the interaction of METTL14 with SMN (new **Appendix Figure S2**). However, PRMT5-catalyzed SDMA modification might contribute more significantly to METTL14–SMN interaction when PRMT1's activity is inhibited, likely due to substrate scavenge (**Fig. 2F** and **2G**). To further support this conclusion, we added additional data showing that while PRMT5 inhibition alone does not significantly impact m⁶A deposition or DNA repair gene expression, its effect becomes pronounced when Type I PRMT activity is inhibited by MS023, as evidenced by the synergistic reduction observed with dual inhibitor treatment (new **Appendix Fig. S3D** and **S3E**).

2. A second major issue is that SMN is involved in splicing. So, knock downs or the use of patient-derived mutants may affect splicing and thus also the m⁶A landscape. Thus, most data on m⁶A levels in this manuscript could be indirectly caused by altering the splicing process.

We thank the reviewer for this insightful suggestion. We agree that the global reduction of mRNA m⁶A levels upon SMN knockdown in HeLa cells and in SMA-patient derived fibroblasts could potentially be an indirect consequence of altered RNA splicing. In the revised manuscript, we added a new discussion section to acknowledge this possibility. Importantly, we also added new data to show that the SMN-mediated regulation of m⁶A deposition on selected DNA repair gene transcripts are unlikely a direct consequence of altered RNA splicing. As shown in new **Appendix Fig. S3C**, the splicing patterns of BLM, ATRIP, and FANCM, were not affected by SMN knockdown.

3. The RGG repeats in METTL14 have been shown to affect RNA substrate binding in a number of publications before. This is largely ignored and at least some of the observed interactions could be due to a loss of RNA interaction of METTL14 and thus interactions could be indirect and mediated through RNA.

We thank the reviewer for this valuable suggestion. We now added new data supporting that although the interaction of METTL14 with SMN is not dependent on RNA (new **Appendix Fig. S1A**), the RNA-mediated interaction does contribute to the residual METTL14–RNAPII association after SMN knockdown (New **Fig. 3C**). Together, these results are consistent with our previous report that arginine methylation promotes METTL14 protein–protein interaction with RNAPII and protein–RNA interaction with its substrates.

4. Figure 1: SMN is nuclear and cytoplasmic. Is there a colocalization of the two proteins in IF? Is SMN binding in the context of its large SMN complex or rather as individual protein? In Figure 1B, METTL3 does not bind to SMN. However, at least a fraction of the tagged protein might form a complex with endogenous METTL14 and thus a complete absence of interaction might be rather unexpected.

We thank the reviewer for this question. As the reviewer suggested, we performed immunofluorescence staining and detected the subcellular co-localization of SMN and METTL14. Indeed, SMN localizes both in the cytoplasm and nucleus, whereas majority of METTL14 localizes in the nucleus. As shown in new **Appendix Fig. S1B**, the two proteins colocalize in the nucleus, with strong signals likely at Cajal bodies, where SMN is known to localize.

We agree with the reviewer's comments that Flag-METTL3 might also pull down SMN through its interaction with endogenous METTL14, and this is indeed the case. We repeated the experiment and included a longer exposure time for Western blot detection. Both Flag-METTL3 and Flag-WTAP were able to co-immunoprecipitate SMN, although to a much lesser extent compared to Flag-METTL14 (new **Fig. 1B**), suggesting that SMN is associated with the methyltransferase complex, likely through METTL14.

5. Figure 2B: binding is already fully abolished when 5 arginines are mutated. Are the other 8 not methylated in this mutant?

We thank the reviewer for this question. To address this, we put together some results from our previous publication (EMBO J (2021) 40: e106309; PMID: 33459381). Although the 5 arginines are the major methylation sites identified on METTL14 by mass spectrometry (**Response Figure 2A**), mutations of the 5 arginine residues only partially reduced

Response Figure 2. PRMT1 methylates METTL14 at multiple arginine residues (from previous publication). (A) Arginine methylation sites of METTL14 detected by mass spectrometry. (B) In vitro methylation assay of with WT and various RK mutant METTL14 incubated with

PRMT1 in vitro (**Response Figure 2B**), suggesting that additional arginine residues can be methylated beyond the 5 sites. However, available ADMA antibody was not able to detect residual methylation signals on the 5RK construct, probably because their low abundance, thus below the detection threshold (new **Fig. 2B**). We speculate that there might be at least two reasons by which 5RK mutation dramatically reduces METTL14 interaction with SMN Tudor domain: 1) the level of residual methylation is low and the interaction of METTL14 5RK with Tudor domain is too weak to be detected by GST pull-down; or 2) SMN Tudor domain mainly interacts with those 5 methylated arginine residues. Notably, these 5 arginine residues are located at the very end of METTL14 C-terminus. We believe it is unlikely that the 5RK mutation would completely abolish METTL14 methylation in cells.

6. Figure 2D: the authors mention that scavenger methylation may occur, when PRMT1 is knocked down, which could explain the interaction with SMN since PRMT5 could now be recruited and generate sDMA. One would expect an increase in SMN binding in such a model. However, a decrease is observed. Similarly, in 2G, a clear effect on interaction is only observed when aDMA and sDMA generation is inhibited, which is not consistent with the strong effect that PRMT1 knock down has. This remains unclear and is unfortunately not followed up further.

We apologize for the confusion and hope that the new data shown in **Response Figure 1** could provide further clarification that SMN Tudor does not specifically or limited to bind SDMA modified substrates. In this study, we believe that ADMA catalyzed by PRMT1 is the major driver for METTL14–SMN interaction. We reason that the scavenger SDMA modification catalyzed by PRMT5 occurs when PRMT1 activity is inhibited and contributes to the remaining METTL14–SMN interaction. It is likely that SDMA modification of METTL14, although being induced under PRMT1 inhibition, does not reach the level of ADMA in control cells. Thus, the net effect of PRMT1 inhibition led to a reduced METTL14–SMN interaction, which is further reduced with dual inhibitor treatment presumably by blocking the scavenger effect.

7. Figure 3 is not convincing. The authors arrive at a model in which SMN mediates the interaction of METTL14 with the CTD of pol II. SMN interacts with the methylated CTD and also interacts with methylated DMA on METTL14. The Tudor domain could only interact with the one

or the other at a time? So how would such a model work? Is there any RNA-dependence in these assays?

We thank the reviewer for raising this very important question. We agree that the working model is somewhat puzzling and not yet fully supported by direct evidence. However, a similar mechanism of action has been observed for other methylarginine effector proteins, such as TDRD3. For instance, the Tudor domain of TDRD3 can recognize histone methylation and recruit DHX9 both in an arginine methylation-dependent manner to regulate transcription (Nucleic Acids Res 2021, PMID 34329467). We are actively investigating this seemingly shared behavior of methylarginine readers by testing the hypothesis that the ability to multimerize or form biomolecular condensates might underlie their mechanism of action. A few recent studies have reported the role of arginine methylation and Tudor domain in condensate formation in vivo (Cell 2021, PMID 34115980; Life Sci Alliance 2022, PMID 36375840; Dev Cell 2024, PMID: 39029469). We were able to detect strong homotypic interaction of SMN with its Tudor domain in GST pull down (**Response Figure. 3**). Thus, it is possible that a multimerized SMN protein complex would be able to mediate multiple arginine methylated substrates for interaction.

Figure for referee with unpublished data and its description has been removed upon request by the authors.

Regarding RNA dependence, our new data showed that RNA is not involved in the interaction of SMN with METTL14 (new **Appendix Fig. S1A**), but the METTL14–RNAPII interaction is facilitated by RNA (new **Fig. 3C**). As RNA is a critical regulator of condensate formation, we agree that further testing is needed to fully determine the contribution of RNA to SMN-mediated METTL14–RNAPII interactions.

8. Figure 4: as mentioned, the rather mild effects that are observed could be due to splicing deficits caused by compromising SMN function and may not be linked to methylation of METTL14. The mouse that has been generated does also not provide further evidence that there is a functional connection.

We agree with the reviewer that the global reduction of mRNA m⁶A levels upon SMN knockdown in HeLa cells and in SMA-patient derived fibroblasts could potentially be an indirect consequence of altered RNA splicing. Although it is challenging to clearly separate SMN's function in splicing regulation and m⁶A homeostasis at transcriptome level, we added new data to show that the SMN-mediated

Figure for referee with unpublished data and its description has been removed upon request by the authors.

regulation of m⁶A deposition on selected DNA repair gene transcripts are unlikely a direct result of splicing alternation. As shown in new **Appendix Fig. S3C**, the splicing patterns of BLM, ATRIP, and FANCM, were not affected by SMN knockdown. We also added a new discussion section to acknowledge this possibility in the revised manuscript.

The interplay between RNA splicing and m⁶A homeostasis is complex. As shown in **Response Figure 4**, knockdown of different splicing factors has various effects on mRNA m⁶A levels, with SF3B1 depletion causing a dramatic m⁶A increase. In our manuscript, we observed a 15-20% reduction in mRNA m⁶A levels upon SMN knockdown in HeLa cells and in fibroblasts from clinically affected SMA patients (**Fig. 4**), comparable to the extent of m⁶A reduction seen in METTL14 knockdown cells (**Response Figure 4**). We believe this finding is significant as it offers new insight into the molecular underpinnings of SMA pathology.

While the most prominent effects of SMN deficiency are observed in the nervous system, there is increasing evidence suggesting that SMN's ubiquitous expression and crucial role in fundamental cellular processes could have broader implications for other tissues, including the hematopoietic system (Hum Mol Genet. 2017, PMID 28062667). Further characterization of METTL14 RK mutant mouse model may reveal more functional connections with SMA.

Referee #2:

In this manuscript by Zhang et al the authors built on their previous discovery regarding the function of methylated arginines in METTL14. Their previously shown that this methylation is important for m⁶A deposition, in particular on transcripts involved in DNA repair. Here they found that SMN can interact with METTL14 through its arginine methylation domain, and consequently this stimulates the function of METTL14. Lack of SMN leads to reduced m⁶A levels in HeLa cells, likely due to the decreased association with RNA Pol II. Interestingly SMN patient derived fibroblasts also display reduction of m⁶A. Lastly they engineered a METTL14 arginine methylation deficient mouse model, showing some impact on embryonic development, alteration of hematopoietic lineage and increased genome instability.

This study uncovers a novel interaction of METTL14 with SMN, which plays a critical role in spinal muscular atrophy. This provides another mechanism by which SMN alteration might contribute to the disease. The manuscript is well written and the experimental data are solid and properly interpreted.

I have only some minor comments that could help further improving the manuscript.

We thank the reviewer for the very positive comments about our work. Our responses to each of the reviewer's comments can be found below.

Figure 4:

- Panel B is not very informative. The decrease m⁶A level is unclear in the SMN KD.
- Panel D should be quantified.

We thank the reviewer for these suggestions. We repeated the m⁶A dot-blot and provided a revised figure demonstrating the clear reduction of m⁶A in SMN knockdown cells (new **Fig. 4B**). We acknowledge that although the m⁶A dot blot is semi-quantitative, our m⁶A LC/MS consistently detected ~15-20% mRNA m⁶A reduction upon SMN loss of function.

As the reviewer suggested, we quantified the SMN Western blot results (updated **Fig. 4D**).

Figure 5

- Panel D: A similar experiment should be repeated with a METTL3 depletion (KD or inhibitor). Otherwise it remains unclear that the effect of SMN depletion on FANCD2 level depends on its m⁶A stimulating activity.

We agree with the reviewer's suggestion. We now added new data showing that the induction of FANCD2 expression upon DNA damage is significantly dampened by either inhibition of m⁶A methyltransferase activity using a small molecule inhibitor targeting METTL3 (STM2457) or knockdown METTL14 using siRNA (new **Appendix Fig. S4A and S4B**), thus supporting an m⁶A-dependent mechanism.

- Panel G: Given the data presented in panels D-F it would be logical to also look at FANCD2 m⁶A and protein levels in the patients derived cells.

As the reviewer suggested, we now added new data showing that the induction of FANCD2 mRNA m⁶A and protein expression upon DNA damage is significantly dampened in fibroblasts from clinically affected SMA patients (new **Appendix Fig. S4E – S4G**).

Figure 6

- Does the level of m⁶A altered in the RK/RK animals as it would be expected?

As the reviewer suggested, we perform m⁶A dot-blot and confirmed that the mRNA m⁶A levels in the spleen, thymus, and bone marrow of RK/RK mice are significantly lower than in WT controls (new **Appendix Fig. S5C and Fig. S6F**).

Referee #3:

The manuscript by Zhang et al explores a new link between arginine methylation and RNA m⁶A homeostasis.

The paper is well written, experiments are carefully controlled, and overall, I am very positive to see it published in EMBORep.

Some suggestions for improvements.

We thank the reviewer for the positive comments about our work. Our detailed responses to each of the reviewer's comments can be found below.

Major.

Fig.2D/E, have the authors tried the same experiment with siPRMT5?

We thank the reviewer for this question. As the reviewer suggested, we now added new data showing that compared to PRMT1, PRMT5 only weakly methylates METTL14 in vitro (new **Appendix Fig. S2A**), and that knockdown PRMT5 in HeLa cells does not significantly affect SMN interaction with METTL14 (new **Appendix Fig. S2B**). We reason that the scavenger SDMA modification catalyzed by PRMT5 occurs when PRMT1 activity is inhibited and contributes to the remaining METTL14–SMN interaction.

Fig.5. Given the published link between PRMT5 and DDR genes, have the authors tried to perform the experiment in Fig.5A and Fig.5C in the presence of EPZ, but more importantly of EPZ+MS023?

We thank the reviewer for this insightful question. Given the established role of PRMT5 in regulating DNA damage response genes, we tested the effects of EPZ alone and in combination with MS023 on m⁶A deposition and DNA repair gene expression. As shown in the newly added **Appendix Fig. S3D** and **S3E**, EPZ treatment alone had minimal effect, but co-treatment with MS023 resulted in a significant synergistic reduction in both m⁶A levels and the protein expression of selected DNA repair genes, supporting a compensatory role for PRMT5 upon type I PRMT inhibition.

Fig.6B. Have the author tried to quantify MMA or SDMA? It would be important to understand if in vivo those sites are mainly ADMA or else.

As suggested by the reviewer, we have added new data showing that, using commercially available pan-MMA and pan-SDMA antibodies, we were unable to detect MMA or SDMA modifications on METTL14 immunoprecipitated from mouse spleen and thymus tissues (new **Appendix Fig. S5B**).

Fig.6. Was the increase in gammaH2AX also observed in hematopoietic cells?

As suggested by the reviewer, we have added new data showing that there is a modest increase in gamma H2AX levels in the bone marrow of METTL14 RK mutant mice compared to wild type controls (new **Appendix Fig. S6G**).

Fig.6. Given the lack of phenotype at steady state, but potential defects in HSCs, have the authors tried a HSC reconstitution in lethally irradiated mice? The stress imposed to HSCs might uncover additional phenotypes.

We thank the reviewer for this insightful question and agree that transplantation may impose additional stress, potentially unmaking phenotypes not apparent under steady-state conditions. Following this suggestion, we conducted HSC reconstitution assays by transplanting CD45.2 WT or CD45.2 Mettl14 RK HSCs into lethally irradiated CD45.1 recipient mice. As shown in new **Appendix Fig. S7**, Mettl14 RK significantly impaired self-renewal capacity of CD45.2 HSCs compared to wild type controls. Moreover, Mettl14 hypomethylation significantly promoted myeloid differentiation, while suppressing B cell development in the bone marrow. This is consistent with our findings in Mettl14 RK knock-in mice.

Minor.

Please fix some spelling mistakes, and some other minor text issues such as:

1. "knockdown of PRMT1 expression using siRNA significantly reduced the interaction of METT14 with SMN as revealed by both co-IP and GST pull down assays (Figure 2D and 2E)." METT14, instead of METTL14,

As the reviewer suggested, we corrected the mis-spelling.

2. "consistent with the general observation of PRMT substrate scavenging (Refs)," Actually add the references.

As the reviewer suggested, we added the reference paper.

3. Were the pulldowns described in Fig.1/2 performed in the presence of RNase?

We thank the reviewer for this question. RNase A treatment was specifically addressed in new **Appendix Fig. S1A** and in new **Fig. 3C**.

Dear Dr. Yang,

Thank you for the submission of your revised manuscript. We have now received the enclosed reports from the referees and I am happy to say that all support its publication now. Only a few editorial requests will need to be addressed before we can proceed with the official acceptance of your manuscript:

- Please add up to 5 keywords to your ms file.
- The conflict of interest subheading needs to be renamed to "Disclosure and Competing Interests Statement"
- There is one author name discrepancy: Hoang Quoc Hai Pham in the ms vs. Hai Pham in the online submission system, please correct.
- The author credits need to be removed from the ms file. All credits need to be entered during online ms submission.
- DATA NOT SHOWN on page 9 is not allowed by our journal policy. Please re-write or include the data.
- The callout for Appendix Table 1 is missing "S", it should be Appendix Table S1. Table EV1 is called out in the Reagents table, but there only is an Appendix Table S1. Please correct. The Appendix Table S1 needs to be part of the Appendix file and listed in the table of content.
- Materials and Methods should be just Methods.
- Please provide exact p values (as reasonable) in the legends of figures 5A, B, E, F, I; 6D, E, F, G.

I would like to suggest some minor changes to the second half of the abstract. Please let me know whether you agree with the following:

Both SMN knockdown and SMA mutations impair m6A deposition on the mRNAs of DNA repair genes [OK?], mirroring the effects of METTL14 hypomethylation. Also, SMA patient fibroblasts are hypersensitive to DNA-damaging agents due to reduced levels of DNA repair gene expression. To explore the function of METTL14 arginine methylation in vivo, we generated a Mettl14 methylation-deficient mouse model (Mettl14RK). Although this model does not show SMA-like phenotypes, the mutants are partially embryonic lethal and show abnormal hematopoiesis, underscoring a role for methylated METTL14 in early development.

EMBO press papers are accompanied online by A) a short (1-2 sentences) summary of the findings and their significance, B) 2-3 bullet points highlighting key results and C) a synopsis image that is exactly 550 pixels wide and 200-600 pixels high (the height is variable). The synopsis image should provide a sketch of the major findings, like a graphical abstract. Please note that text needs to be readable at the final size. Please send us this information along with the final manuscript.

Referee #1:

The authors have revised their manuscript and addressed all points that I had raised on the previous version. Generally, the authors have adequately responded to the issues raised. New experiments have been performed and several aspects became clearer now.

Although the general model and particularly the link to regulation of m6A modification is still rather preliminary and requires solidification, the manuscript may inspire research into this novel direction. SMN is part of a large protein complex and forms a molecular machine that evolved to load Sm protein rings onto snRNAs. It would still be informative to know whether SMN acts

outside this macromolecular complex or as part of it, when binding to the m6A writer complex.

I am nevertheless satisfied with the response to my comments.

Referee #2:

The authors adequately addressed all my comments and I recommend the publication of this interesting piece of data.

Referee #3:

I thank the authors for the effort. They have addressed all the relevant point raised and went beyond my expectations in revising this manuscript with compelling new data.

Response to editorial decision letter

Thank you for the submission of your revised manuscript. We have now received the enclosed reports from the referees, and I am happy to say that all support its publication now. Only a few editorial requests will need to be addressed before we can proceed with the official acceptance of your manuscript:

- Please add up to 5 keywords to your ms file.

We provided 5 keywords for our manuscript.

- The conflict of interest subheading needs to be renamed to "Disclosure and Competing Interests Statement"

We changed the subheading according to the suggestion.

- There is one author name discrepancy: Hoang Quoc Hai Pham in the ms vs. Hai Pham in the online submission system, please correct.

We changed the author's name to "Hoang Quoc Hai Pham" in the online submission system.

- The author credits need to be removed from the ms file. All credits need to be entered during online ms submission.

We removed the author contribution section from the manuscript.

- DATA NOT SHOWN on page 9 is not allowed by our journal policy. Please re-write or include the data.

We removed the "data not shown" statement and revised the text accordingly.

- The callout for Appendix Table 1 is missing "S", it should be Appendix Table S1. Table EV1 is called out in the Reagents table, but there only is an Appendix Table S1. Please correct. The Appendix Table S1 needs to be part of the Appendix file and listed in the table of content.

We corrected the callout for "Appendix Table S1".

Table EV1 is changed to "Appendix Table S1" in the Reagents table.

We combined Appendix Table S1 with the Appendix Figures into one Appendix file.

- Materials and Methods should be just Methods.

We changed the heading to "Methods" according to the suggestion.

- Please provide exact p values (as reasonable) in the legends of figures 5A, B, E, F, I; 6D, E, F, G.

We added exact p values to above mentioned figure legends. We also updated Figure 5 accordingly.

I would like to suggest some minor changes to the second half of the abstract. Please let me know whether you agree with the following:

Both SMN knockdown and SMA mutations impair m6A deposition on the mRNAs of DNA repair genes [OK?], mirroring the effects of METTL14 hypomethylation. Also, SMA patient fibroblasts are hypersensitive to DNA-damaging agents due to reduced levels of DNA repair gene expression. To explore the function of METTL14 arginine methylation in vivo, we generated a Mettl14 methylation-deficient mouse model (Mettl14RK). Although this model does not show SMA-like phenotypes, the mutants are partially embryonic lethal and show abnormal hematopoiesis, underscoring a role for methylated METTL14 in early development.

Thanks for the suggestions. We have revised the abstract according to your suggestion. It reads much clear now.

EMBO press papers are accompanied online by A) a short (1-2 sentences) summary of the findings and their significance, B) 2-3 bullet points highlighting key results and C) a synopsis image that is exactly 550 pixels wide and 200-600 pixels high (the height is variable). The synopsis image should provide a sketch of the major findings, like a graphical abstract. Please note that text needs to be readable at the final size. Please send us this information along with the final manuscript.

We now included a short summary, bullet points, and a graphic abstract in the package.

I look forward to seeing a final version of your manuscript as soon as possible. Please use this link to submit your revision: <https://embor.msubmit.net/cgi-bin/main.plex>

Dr. Yanzhong Yang
Beckman Research Institute, City of Hope
Department of Cancer Genetics and Epigenetics
1500 E. Duarte Road
Duarte, California 91010
United States

Dear Dr. Yang,

I am very pleased to accept your manuscript for publication in the next available issue of EMBO reports. Thank you for your contribution to our journal.
